# Tad pili with adaptable tips mediate contact-dependent killing during bacterial predation

Julien Herrou [1] ✉, Laetitia My [1], Caroline L. Monteil [2], Marine Bergot[2], Rikesh Jain [1], Emmanuelle Martinez[1] & Tâm Mignot [1] ✉

The predatory bacterium, *Myxococcus xanthus*, kills its prey by contact, using a putative Tight Adherence pilus, known as the Kil system, along with a protein complex resembling the basal body a type-III secretion system, named the "needleless" T3SS*. In this work, we provide direct evidence that *Myxococcus* polymerizes a Kil pilus at the prey contact site, which is constituted by the major pilin KilP. We also genetically demonstrate that the predation function of this pilus is linked to four different minor pilin complexes, which work in specific combinations to detect and kill phylogenetically diverse bacterial species. Structural models of the Kil pilus suggest that these minor pilin complexes form interchangeable "Tips", exposing variable domains at the extremity of the pilus to interact with prey cells. Remarkably, the activity of these Tips also depends on the T3SS*, revealing a tight functional connection between the Kil system and the T3SS*. While these Tips are mostly restricted to predatory bacteria, genomic and structural analyses suggest that in other bacteria, including pathogens, Tad pili are also customized and functionalized by similar minor pilin complexes exposing variable domains.

Bacteria interact with their environment with a variety of extracellular filamentous appendages (e.g., flagella, pili, fimbriae, secretion, conjugative systems, etc.), allowing them to colonize an incredible diversity of ecological niches[1–7]. Among these, the Type-IV pili (T4P) superfamily regroups a very ancient and widespread ensemble of filamentous nanomachines, including the T4aP, T4bP, T4cP/Tad, Com, and T2SS subclasses. These systems have evolved to perform a diversity of functions, such as adhesion, biofilm formation, motility, effector secretion, and DNA uptake, to name a few[4,5,8–13].

In diderm bacteria, these machineries enable the polymerization of a pilus, which is powered by an inner membrane (IM) ATPase-containing complex and exported, through the outer membrane (OM), by a secretin[4,10,12,14]. The pilus itself is primarily composed of multiple copies of a protein subunit known as the major pilin. Pilins adopt a classic lollipop-like shape, featuring a conserved N-terminal α-helix (α1) and a variable C-terminal β-stranded globular domain[4,10,11,15]. Within a pilus, thousands of major pilin monomers are assembled. Their N-terminal regions form tightly packed α-helical

bundles, which constitute the backbone of the filament. In contrast, their globular domains are solvent-exposed, conferring electrostatic surface properties to the pilus. Typically, pilin subunits are secreted in the periplasm via the general secretory pathway. These pilins are next processed in the IM by a prepilin peptidase before their subsequent incorporation into the polymerizing fiber[4,12,16]. Pilus polymerization is initiated by so-called minor pilins, which constitute the pilus tip. Like most pilins, minor pilins also consist of an α1-helix and a globular head but often feature additional domains that are exposed at the pilus end, conferring specific properties[10,15,17–20]. For instance, the tip of *Streptococcus sanguinis* T4aP is equipped with a heterotrimeric complex of minor pilins named PilA, PilB, and PilC[21]. PilC and PilB feature a lectin domain and a von Willebrand factor A (VWA) domain, which confer adhesion properties to the pilus, enabling its interaction with the extracellular matrix[21,22]. Minor pilins can also contribute to the delivery of virulence factors or effectors. In *Vibrio cholerae*, the terminal minor pilin TcpB docks TcpF, a key factor in host colonization[23]. Similarly, type-II secretion systems

[1]Aix Marseille Univ, CNRS, LCB, Marseille, France. [2]Aix-Marseille Univ, CEA, CNRS, BIAM, Saint-Paul-lez-Durance, France. ✉e-mail: jherrou@imm.cnrs.fr; tmignot@imm.cnrs.fr

(T2SS) use pilus-associated minor pilins to load and deliver toxins and enzymes into the environment[24,25].

More recently, it has been shown that T4Ps also play a critical role in bacterial predation[26–28]. *Myxococcus xanthus* is a ubiquitous soil bacterium that moves collectively and kills its prey by contact. Two distinct T4Ps contribute to this predation process: (i) a twitching pilus (T4aP) functions as a motility engine, enabling group movements and facilitating prey colony invasion by *Myxococcus*[19,27,29–32], (ii) while a Tad-like pilus (T4cP), known as the Kil system, is essential for the contact-dependent killing of the prey during predation[27,28]. In bacteria, Tad pili plays an important role in diverse biological processes such as adhesion, biofilm formation, surface sensing, virulence, DNA uptake, or twitching motility[33,34]. The identification of the Kil system expands the functional repertoire of Tad systems, highlighting the extensive diversification of these machineries[27,33]. Remarkably, the Kil system exhibits broad-spectrum killing capabilities against a wide range of phylogenetically distant prey, including both monoderm and diderm bacteria (e.g., *Escherichia coli*, *Caulobacter crescentus*, *Bacillus subtilis*, *Salmonella enterica*, etc.), suggesting the implication of a large array of toxic effectors[27].

In *Myxococcus*, the Kil system is encoded by the *kil*-gene clusters I and II (*mxan_3102-3108* and *mxan_4648-4661*). While many of these genes have no identified functions, all the predicted core Tad components−including the secretin KilC, the CpaB homolog KilB, the IM platform proteins KilGH and the ATPase KilF−are essential for predation[27]. In addition, fluorescent fusions of KilC, KilH, and KilF form bright and dynamic clusters at the contact site between *Myxococcus* and the prey, indicating that the Kil system is recruited and assembled at this location upon contact[27,28]. Remarkably, this coincides with motility arrest, followed by the death of the prey within minutes. This suggests that assembly of the Kil system is dynamically regulated by prey cell contact, and coupled to motility regulation and prey intoxication[27].

A second apparatus, resembling a type-III secretion system (T3SS), is also recruited at the prey contact site. This system is encoded by a large gene cluster (*mxan_2434-2464*) containing all the components needed for the assembly of a T3SS-like sorting platform but lacking key elements, such as secretin, needle, and translocon, essential for the delivery of toxic effectors in eukaryotic cells[28,35–39]. This "needleless" T3SS* is not required for prey killing but is essential for prey lysis, a crucial step in accessing prey nutrients[28]. Recruitment of the Kil and T3SS* appears interdependent[28] (revisited herein), suggesting that these two systems function together as predation machinery.

Understanding the predation process requires elucidating the distinct contributions of these systems. While genetic evidence points to a "Tad pilus"-like function for the Kil apparatus, the role of its pilins remains unclear. Specifically, the *kil*-gene cluster II encodes two predicted minor pilins (KilL and KilM), one potential major pilin (KilK), and a prepilin peptidase (KilA)[27]. However, the deletion of *kilK* only resulted in a partial predation phenotype, and the Δ*kilA* mutant showed no noticeable predation defect[27]. This is surprising, as major pilins and prepilin peptidases are typically essential for the assembly of T4Ps[4,10,11,14]. Hence, it is possible that the Kil system does not work as a canonical Tad pilus, or that the Kil major pilin is still missing.

In this work, we aimed to clarify the function of the Kil apparatus and its interaction with the T3SS* during predation. We determined that the full Kil-pilin repertoire consists of 14 minor pilins (KilK, KilL$_{1-4}$, KilM$_{1-4}$, KilN$_{1-4}$, and KilO) and one major pilin (KilP), and we demonstrated that the Kil apparatus polymerizes a pilus fiber, which is essential for perceiving and killing prey cells. Remarkably, the extremity of the Kil pilus can be functionalized by up to four distinct minor pilin complexes, referred to as "Tips". The prepilin peptidase KilA is essential for the maturation of a subset of these Tips. Moreover, analysis of single-Tip expressing mutants revealed varying predation efficacies depending on the prey species, suggesting that the Kil machinery adapts its Tip composition to the type of prey. Importantly, all four Tips are functionally associated with the T3SS*, reflecting a possible association between these Tips and potential T3SS* effectors. Finally, AlphaFold[40,41] models provided insights into the structural arrangement of each Tip at the Kil pilus extremity and suggested that Tad pili are similarly functionalized by variable Tip-like complexes in other bacteria.

## Results

### *Myxococcus* polymerizes an Flp pilus at the prey contact site

Previously, we identified KilK (*mxan_4655*) as the potential major pilin of the Kil system. However, a *kilK* deletion mutant showed only a modest decrease in its predation efficiency[27]. This was intriguing, as major pilins are considered to be essential components of bacterial pili[4,10,11,15]. This either suggested that the Kil system functioned differently or that the Kil major pilin had yet to be identified.

We therefore looked for additional pilin candidates that may have been ignored in our original study. To investigate this, we searched the *Myxococcus* genome for genes encoding Tad-like pilins, also referred to as Fimbrial low-molecular-weight proteins (Flp). Flps display unique characteristics among Type-IV pilins: they are short proteins (ranging from 50 to 80 amino acid residues) that adopt a fully α-helical conformation. Like other pilins, Flps are initially expressed as prepilins and are subsequently cleaved by a prepilin peptidase at a conserved N-terminal cleavage motif[4,12,15,26,33,42–47]. Based on these characteristics, we identified a gene (*mxan_5121*), conserved among the *Myxococcales*, and encoding a 97-amino acid long protein. This protein presents a predicted leader peptide region followed by a cleavage motif consisting of a conserved glycine immediately preceding the cleavage site and separated from invariant glutamate by four residues (G/XXXXE)[10] (Supplementary Fig. 1a). AlphaFold[40,41,48] predicted with high confidence that this protein is fully α-helical, with a distinctive N-terminal hydrophobic region (α1N) present after the conserved cleavage motif (Supplementary Fig. 1b). Hence, this protein exhibits all the sequence and structural characteristics of bona fide Flps[5,46,49]. In the rest of the manuscript, we will refer to this protein as KilP (Kil-system major Pilin).

To test the importance of KilP in predation, we first evaluated the predation efficiency of a *kilP* deletion mutant. Similar to a Δ*kilACF* mutant (Δ*mxan_3105-3107*), which cannot assemble the major structural components of the Kil system[27], a Δ*kilP* mutant was fully impaired in its ability to predate *E. coli* on agar plates (Fig. 1a). This phenotype was also confirmed in a predation liquid assay, which measures the release in the supernatant of cytoplasmic β-Galactosidase by *E. coli* when lysed by *Myxococcus*[27] (Supplementary Fig. 1c). The Δ*kilP* predation defect was complemented by the ectopic expression of KilP, demonstrating its importance in predation (Supplementary Fig. 1d, e).

Next, to test whether *Myxococcus* polymerizes a pilus at the prey contact site, we adapted a pilus-labeling method using a thiol-reactive maleimide dye (AlexaFluor-488-C$_5$ maleimide)[50–53]. To label KilP, we substituted its alanine at position 96 with a cysteine (A96C) (Supplementary Fig. 1a). This variant (KilP$^{A96C}$) was then ectopically and constitutively expressed in a Δ*kilP* strain. However, when expressed alone, KilP$^{A96C}$ failed to complement this mutant, indicating that the protein was not functional for predation (Supplementary Fig. 1d, e). To address this issue, KilP$^{A96C}$ was expressed in a wild-type strain. When this merodiploid strain was exposed to *E. coli*, the KilP$^{A96C}$ pilin formed fluorescent foci at the prey contact site, and this was followed by the polymerization of a fluorescent pilus extending towards the prey and retracting subsequently (Fig. 1b and Supplementary Movie 1). Thus, fluorescently labeled KilP$^{A96C}$ monomers can be incorporated into the native pilin filament. Pilus formation correlated with motility pausing and prey cell lysis. We determined that pilus extension became visible one minute after the pause, which was followed by prey lysis within a minute (Supplementary Fig. 1f).

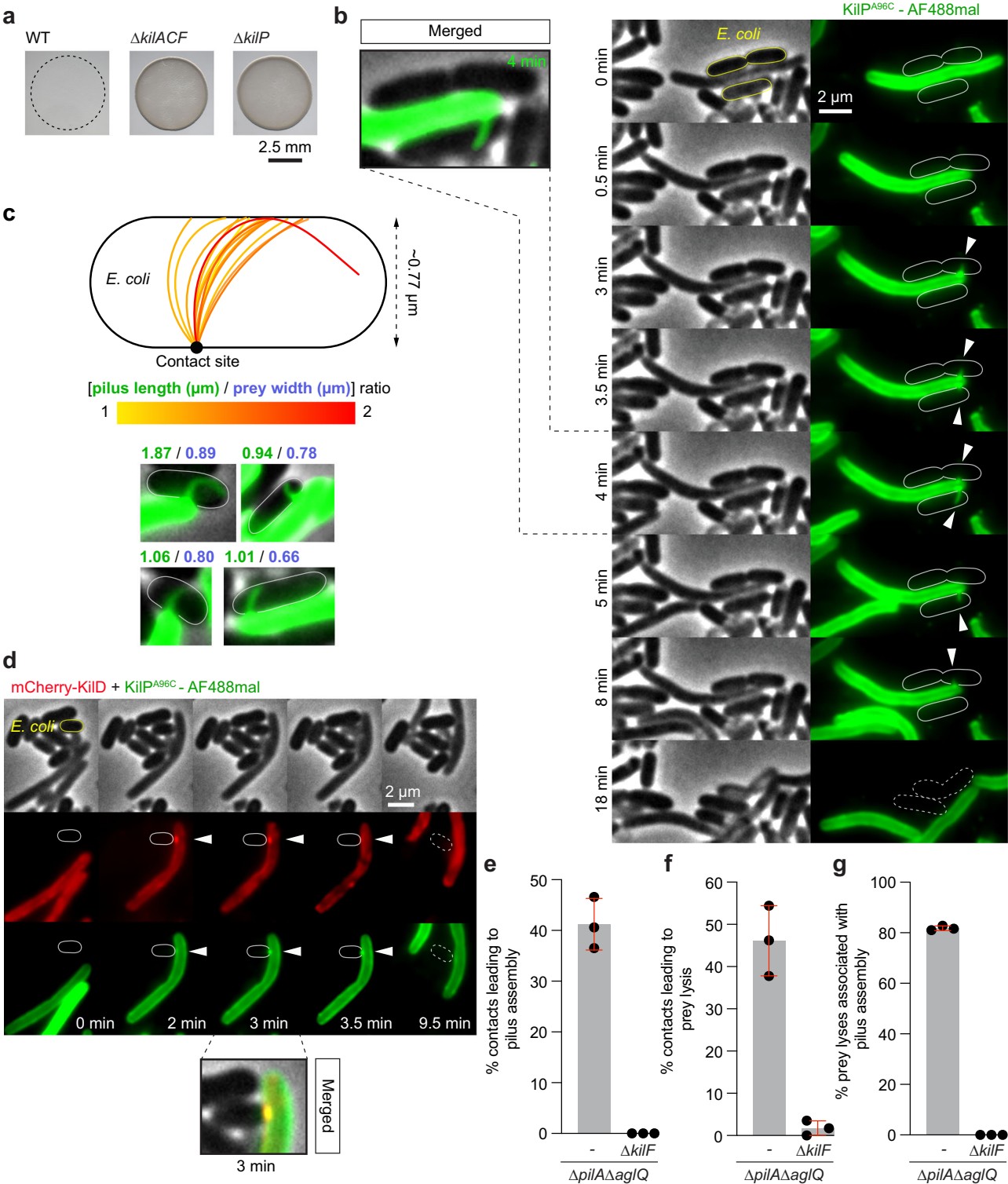

On average, we estimated that the KilP pilus length was $0.4 \pm 0.2\,\mu m$ (Supplementary Fig. 1g). Occasionally, the pilus exceeded the prey cell width ($\sim 0.8\,\mu m$), and in this case, it systematically curved when reaching the opposite side of the prey cell. This suggests that long KilP fibers either wrap around or, more likely, penetrate the prey, bending when they reach the cell membrane on the opposite side (Fig. 1c).

Overall, these results demonstrate that *Myxococcus* can polymerize a KilP pilus, and its polymerization is induced by contact with the prey.

## The KilP pilus polymerization is Kil-dependent

To investigate the role of the Kil system in KilP polymerization, we next tested if the assembly of the Kil complex and polymerization of KilP were coordinated spatiotemporally. For this, we followed the colocalization of the KilP pilus with the cytoplasmic protein KilD, a component of the Kil system that is recruited at the prey contact site[27]. Upon contact with the prey, the formation of mCherry-KilD clusters preceded the accumulation of KilP[A96C] at that same site, consistent with KilP polymerization occurring in a Kil-dependent manner (Fig. 1d). To further demonstrate this, we then monitored pilus formation in a

**Fig. 1 | KilP is the Kil major pilin and its polymerization is Kil-dependent.**
**a** Predation of a Δ*kilP* strain is impaired on solid surfaces. *E. coli* and motile *Myxococcus* wild-type (WT), Δ*kilACF* or Δ*kilP* strains were mixed at an 8:1 prey-to-predator ratio and spotted on CF 1.5% agar plates supplemented with 0.075% glucose. Representative images were captured after 24 h of predation. **b** Microscopy images showing two AF488-maleïmide-labeled KilP$^{A96C}$ pili (white arrowheads) polymerizing at the prey contact site. It correlates with motility pausing and *E. coli* lysis. **c** The KilP pilus bends when its size exceeds *E. coli* width. The upper panel represents superimposed pilus traces (n = 14), color-coded according to the [pilus length/prey width] ratio. The average width of *E. coli* is also indicated. The lower panel shows examples of pili bending upon reaching the opposite side of the prey cell. The corresponding pilus lengths (green) and prey widths (blue) are indicated above each picture. **d** Upon contact with prey, polymerization of the KilP pilus

(green fluorescence, white arrowhead) co-occurs with the recruitment of mCherry-KilD (red fluorescence, white arrowhead). **e** This graph shows the percentage of contacts with *E. coli* leading to KilP$^{A96C}$ polymerization in a non-motile (Δ*pilA*Δ*aglQ*) strain carrying or not a *kilF* ATPase deletion. **f** The percentage of contacts leading to prey lysis was also evaluated in these same strains. **g** Finally, the percentage of prey lyses associated with KilP$^{A96C}$ polymerization was determined as well. In panels **e**, **f**, the number of prey contacts used for quantification was n = 1037 for Δ*pilA*Δ*aglQ* and n = 931 for Δ*pilA*Δ*aglQ*Δ*kilF*. In panel **g**, the number of prey lysis events used for quantification was n = 479 for Δ*pilA*Δ*aglQ* and n = 15 for Δ*pilA*Δ*aglQ*Δ*kilF*. These quantifications were obtained from three independent 30-min microscopy movies. Error bars represent the standard deviation of the mean. Source data are provided as a Source Data file.

mutant lacking KilF (*mxan_3107*), the predicted Kil pilus polymerization / depolymerization ATPase[27,54]. Importantly, since *kil* mutants do not pause upon contact with prey cells[27], pilus detection might be hindered in this mutant. To address this issue, we quantified KilP pilus formation in non-motile (Δ*pilA*Δ*aglQ*) strains incubated with *E. coli*. As expected, non-motile *Myxococcus* could still polymerize a KilP pilus at the prey contact site, and it consistently correlated with prey lysis (Fig. 1e–g and Supplementary Fig. 1h). In contrast, both events were fully abolished in a non-motile strain carrying a Δ*kilF* deletion (Fig. 1e–g).

Previously, we also demonstrated that the Kil system is active against a variety of Gram-positive and Gram-negative bacterial species[27]. Consistent with this, polymerization of KilP$^{A96C}$ was observed upon contact with *B. subtilis* or *C. crescentus*, and it systematically correlated with prey cell death (Supplementary Fig. 2a, b).

Altogether, these results demonstrate that the Kil system drives the KilP major pilin polymerization and that the pilus assembly is linked to prey cell lysis.

### *Myxococcus* possesses four different minor pilin complexes that might constitute the tip of a pilus

The identification of KilP as the Kil major pilin led us to reevaluate the possible function of the predicted pilin-like proteins KilM (*mxan_4660*, now referred to as KilM$_4$ in this manuscript) and KilL (*mxan_4658*, now referred to as KilL$_4$), and to investigate the role of the uncharacterized protein encoded by *mxan_4659* (referred to as KilN$_4$). All three genes are located within the *kil*-gene cluster II[27], and the corresponding proteins are homologous to the TadE, TadF, and TadG minor pilins, which are typically encoded by genes present in *tad* loci. However, the exact function of these pilins and their association with a Tad pilus remain elusive[33,34]. AlphaFold predictions confirmed that KilL$_4$, KilM$_4$ and KilN$_4$ adopt a pilin-like fold, characterized by an N-terminal α1N hydrophobic helix followed by a C-terminal β-stranded globular domain (Fig. 2a). Additionally, each α1-helix contains a predicted prepilin peptidase cleavage site positioned after the leader peptide region (see amino acid sequence alignments in Supplementary Fig. 3). Structurally, KilM$_4$ is the shortest and resembles a bona fide pilin. In contrast, KilL$_4$ and KilN$_4$ present extended C-terminal regions. Notably, KilN$_4$ is a large pilin, harboring an additional domain of unknown function with no structural homology to any characterized protein in the protein data bank (PDB). This domain includes two long α-helices, a large β-scaffold, and additional loops and small α-helices (Fig. 2a).

AlphaFold also confidently predicted that KilL$_4$, KilM$_4$, and KilN$_4$ form a heterotrimeric complex, referred to as Tip4 here (see below) (Fig. 2a and Supplementary Figs. 4, 5). In this complex, the pilin monomers are intricately interwoven, their N-terminal α1-helices forming a tightly packed bundle (Fig. 2a and Supplementary Fig. 6). Specifically, the pilin-like region of KilN$_4$ forms the backbone of this heterotrimeric complex and interacts with KilL$_4$ and KilM$_4$. The additional regions of KilL$_4$ wrap around KilN$_4$ and KilM$_4$, establishing numerous intermolecular contacts with these two proteins as well.

Finally, the domain of unknown function of KilN$_4$ constitutes the extremity of this complex (Fig. 2a and Supplementary Fig. 6).

In summary, this minor pilin complex is shaped like a spearhead, its base formed by a bundle of α1N helices and its extremity by the distal domain of KilN$_4$. This structural organization strongly suggests that these proteins are exposed at the tip of a pilus.

Previously, we reported that the deletion of KilL$_4$ and KilM$_4$ had a minor impact on *Myxococcus* predation efficiency in liquid[27]. This finding is intriguing, as minor pilins are generally known to play a critical role in initiating pilus assembly[10,11,15,17–19]. This led us to speculate that other pilins with redundant functions were involved in predation. Using diverse bioinformatic approaches, we identified three uncharacterized gene clusters in *Myxococcus* genome, each paralogous to *kilL$_4$*, *kilM$_4$*, and *kilN$_4$* (see amino acid sequence alignments in Supplementary Fig. 3). For clarity, these *kiLMN*-like gene clusters are numbered from 1 to 3 and referred to as "Tips" (Fig. 2b). AlphaFold predicted with high confidence that the corresponding pilin-like proteins assembled into three distinct heterotrimeric complexes (Fig. 2b and Supplementary Figs. 4, 5). These Tip1, Tip2, and Tip3 complexes are also shaped like spearheads, structurally resembling Tip4. Notably, the C-terminal domain of the KilN paralogs constitutes the extremity of these complexes, conferring distinct structural and electrostatic properties to the Tips (Supplementary Fig. 7a, b). All these Tips also feature numerous predicted intramolecular disulfide bonds, which likely contribute to the folding and stability of the protein assemblies (Supplementary Figs. 3, 7c). The base of each predicted complex is formed by a conserved bundle of α1N helices, which may interact with the end of a pilus (further discussed below) (Fig. 2b and Supplementary Figs. 3, 8).

In summary, this in silico analysis revealed that the *kil*-gene cluster II encodes a potential Tip4 minor pilin complex, which presents strong structural similarities with three additional predicted minor pilin complexes (Tip1, Tip2, and Tip3), each encoded by distinct *tip*-gene clusters.

### The different minor pilin "Tip" complexes play an essential role in predation

We next tested the hypothesis that the predation function of the KilP pilus was linked to these Tips. For this, we systematically deleted the different *tip*-gene clusters and evaluated their respective genetic contributions to predation on surfaces and in liquid. Remarkably, none of the single-*tip* deletion mutants (Δ*Tip$^n$*) showed a predation defect on agar plates. In addition, only partial predation defects were observed in liquid, with the *tip1*-deletion strain being the most affected (Fig. 3a, b and Supplementary Fig. 9a). These results suggest that the Tips are either weakly associated with the pilus function or they have complementary and partially redundant roles in predation. To investigate this further, we systematically tested the effect of all possible *tip*-deletion combinations on *Myxococcus* predation performance.

First, a mutant lacking all the Tips (Δ*Tip$^{1+2+3+4}$* = Tip$^∅$) was fully impaired in killing its prey on agar plates and in liquid (Fig. 3a, b and

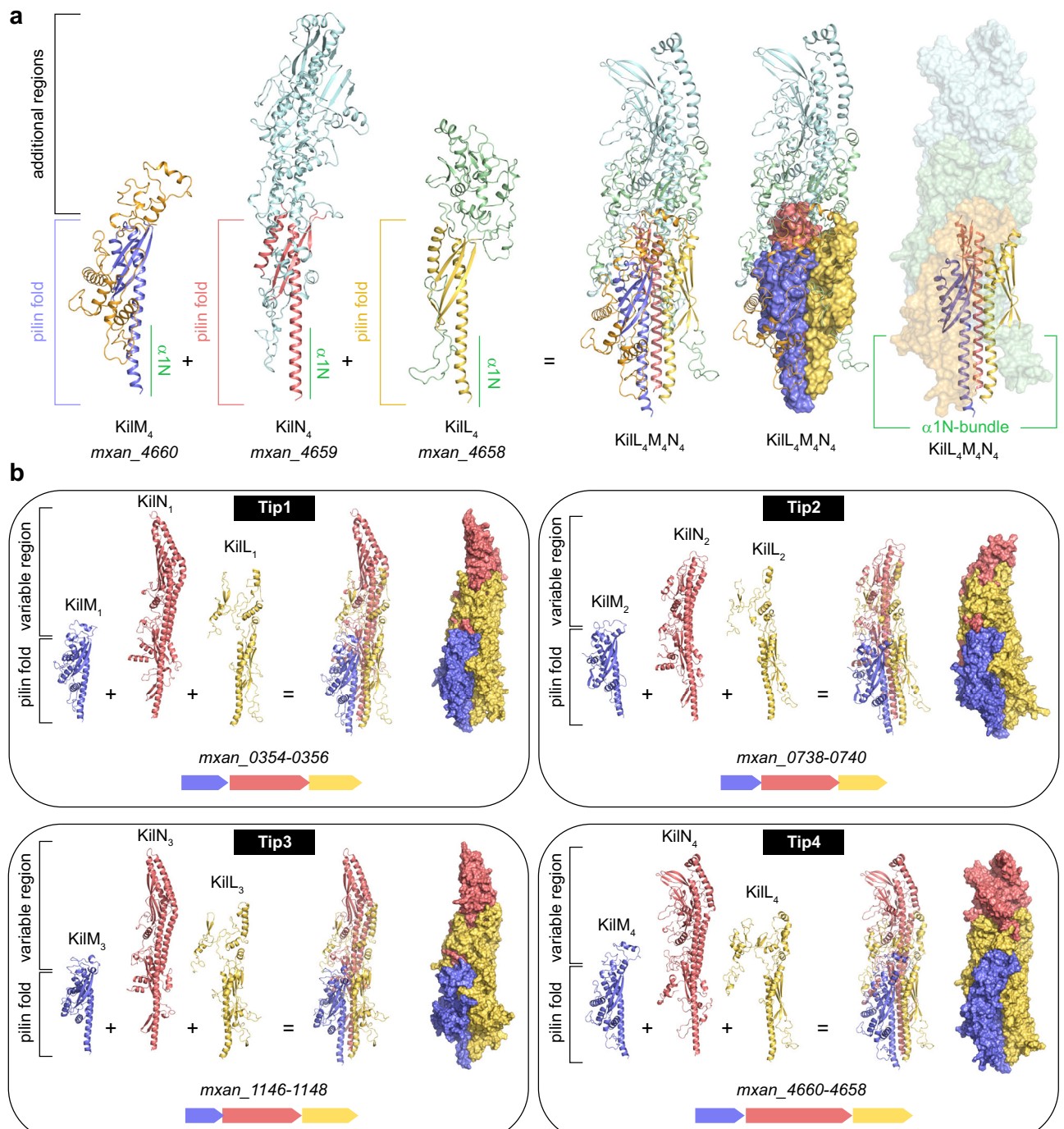

**Fig. 2 | Four distinct KilLMN Tip complexes are present in the *Myxococcus* genome. a** AlphaFold predictions revealed that KilL$_4$, KilM$_4$, and KilN$_4$ adopt a pilin-like fold and feature additional modular domains at their extremities. These proteins assemble to form a heterotrimeric Tip complex, its base is constituted by a bundle of three α1N helices. **b** Four paralogous *kilLMN* gene clusters were identified in the *Myxococcus* genome, each corresponding to a distinct Tip complex. Gene numbers are reported below each structure prediction. To generate these Alpha-Fold predictions, the leader peptide regions were systematically removed from the minor pilin amino acid sequences.

Supplementary Fig. 9a). This indicates that the four *tip*-gene clusters likely encode the entire Kil-minor-pilin repertoire. Next, we tested the individual contribution of each Tip by characterizing strains carrying all possible triple-deletion combinations of the Tips ($\Delta Tip^{x+y+z} = Tip^{(n)}$). In these mutants, the order of importance of the Tips for predation was Tip4 ≥ Tip1 > Tip3 ≥ Tip2 on agar plates (Fig. 3a and Supplementary Fig. 9a), while it was Tip1 > Tip3 > Tip4 > Tip2 in liquid (Fig. 3b). These functional hierarchies were further confirmed by testing all possible double-deletion combinations of the Tips ($\Delta Tip^{x+y}$) (Fig. 3a, b and Supplementary Fig. 9a).

Altogether, these findings demonstrate that, while all the Tips contribute to killing, they perform partially redundant functions in predation. In addition, their efficiencies vary depending on the environmental conditions.

To investigate whether the predation performance of the Tips was related to their expression levels, we also measured their transcription

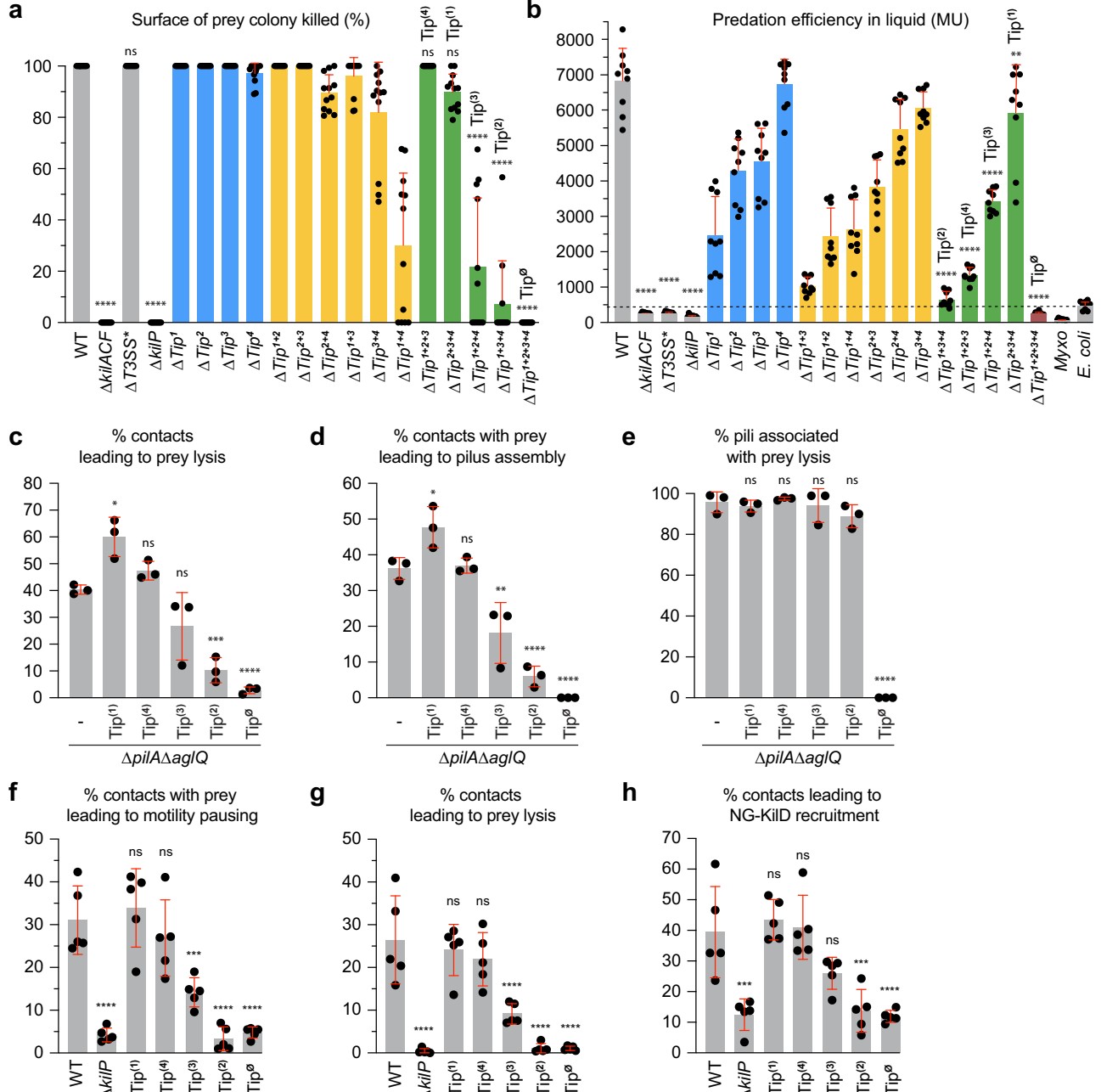

**Fig. 3 | The different minor pilin Tip complexes play an essential role in predation. a** The predation efficiency of the different Tip mutants was quantified by measuring the surface of *E. coli* colony killed by motile *Myxococcus* strains after 24 h of predation. For this assay, an 8:1 prey-to-predator ratio was used. This experiment was performed independently four times on CF agar plates supplemented with 0.075% glucose. Each data point represents a prey colony (*n* = 12 per strain). **b** The predation efficiency of the different Tip mutants was also evaluated in liquid CF after 24 h of incubation. Predation efficiencies in liquid were determined by measuring the kinetics of CPRG hydrolysis catalyzed by the β-Galactosidase released from lysed *E. coli* in the supernatant. *Myxococcus* and *E. coli* alone were used as controls. This experiment was repeated independently three times, and measurements were performed in triplicate (*n* = 9 per strain). MU Miller units. **c** The percentage of contacts leading to *E. coli* lysis was evaluated by microscopy in non-motile (*ΔpilAΔaglQ*) strains expressing all the Tips (-), one Tip only (Tip^(n)), or no Tips (Tip^∅). **d** The percentage of contacts with *E. coli* leading to KilP^A96C polymerization was also evaluated for these strains. In panels **c**, **d**, the number of prey contacts used for quantification was *n* = 1021 for the mutant expressing all Tips, *n* = 983 for Tip^(1), *n* = 1182 for Tip^(2), *n* = 1133 for Tip^(3), *n* = 1007 for Tip^(4), and *n* = 1192

for Tip^∅. **e** The percentage of KilP^A96C pili leading to prey lysis was also determined for these strains. The number of KilP^A96C polymerization events used for quantification was *n* = 370 for the mutant expressing all Tips, *n* = 461 for Tip^(1), *n* = 72 for Tip^(2), *n* = 215 for Tip^(3), *n* = 368 for Tip^(4), and *n* = 0 for Tip^∅. **f** In motile single-Tip expressing strains, the percentage of contacts with prey leading to motility pausing was determined by microscopy and compared to WT, Δ*kilP*, or Tip^∅ strains. **g** The percentage of contacts leading to prey lysis was evaluated for these strains as well. **h** The percentage of contacts with prey leading to NG-KilD recruitment was also determined. In panels **f**–**h**, the number of prey contacts used for quantification was *n* = 1312 for WT, *n* = 2916 for Δ*kilP*, *n* = 1251 for Tip^(1), *n* = 3529 for Tip^(2), *n* = 2005 for Tip^(3), *n* = 1294 for Tip^(4), and *n* = 2614 for Tip^∅. In panels **c**–**e** and **f**–**h**, quantifications were obtained from three and five independent 30-min microscopy movies, respectively. In all panels, error bars represent the standard deviation of the mean. Statistics were determined using ordinary one-way ANOVA followed by Dunnett's multiple comparisons test. Significance compared to strains expressing all Tips: not significant (ns) ^ns*P* > 0.05; significant: **P* < 0.05, ***P* < 0.01, ****P* < 0.001, *****P* < 0.0001. Source data and exact *p* values are provided as a Source Data file.

levels during predation on *E. coli*. We determined that all the *tip*-gene clusters were transcribed, but at significantly lower levels than *kilP*, which is expected for minor pilins[4,10,11,15] (Supplementary Fig. 9b). The transcription levels followed this order: Tip4 > Tip3 > Tip1 > Tip2, suggesting that Tip selection during predation is influenced by their expression levels. However, genetic analyses and microscopy approaches (see below) also revealed a functional link between each of the Tips and both pilus formation and prey killing. Thus, each of the Tip complexes must be expressed at a sufficient level to perform its function.

### The frequency of KilP pilus extension, motility pausing, and prey killing is Tip-dependent

We next aimed to understand how the different Tips present distinct predation efficiencies on *E. coli* cells. Since minor pilins play a role in initiating pilus polymerization[10,11,15,17–19], we evaluated the dynamics of KilP pilus extension in single-Tip expressing strains (i.e., the $\Delta Tip^{x+y}{}^{+z}$ = Tip$^{(n)}$ mutants). Upon contact with *E. coli*, pilus formation was observed in all these strains, and it was systematically associated with motility pausing and prey lysis (Supplementary Fig. 10). Again, to facilitate pilus detection in contact with prey cells, we used non-motile strains expressing a single-Tip complex. In these single-Tip expressing strains, we characterized the contact-dependent killing of *E. coli* and determined that prey lysis efficiency, corresponding to the percentage of contacts leading to *E. coli* lysis, was Tip-dependent. Tip performances for prey lysis followed the same trend previously observed on agar plates, with Tip1 ≥ Tip4 > Tip3 > Tip2 (Fig. 3c). Additionally, pilus polymerization was detected in all these strains, except in a mutant lacking all four Tips (Tip$^{\varnothing}$) (Fig. 3d). Thus, each of the Tip complexes is sufficient to prime pilus extension, which only becomes abolished when all Tips are deleted. Upon prey contact, the frequency of pilus polymerization also varied depending on the Tip, following the same trend as above, with Tip1 ≥ Tip4 > Tip3 > Tip2 (Fig. 3d). Importantly, when a pilus was formed, it was systematically associated with prey lysis in all these strains (Fig. 3e). In conclusion, the Tips show varying efficacies in prey cell recognition, but they all support killing.

To further evaluate the importance of the Tips in prey recognition, we analyzed motile strains expressing single-Tip complexes and tested their ability to induce a pause in contact with prey cells. In these strains, the frequencies of contact-dependent pauses and lyses were also influenced by the type of Tip, following a similar hierarchy, with Tip1 ≥ Tip4 > Tip3 > Tip2 (Fig. 3f, g). Using NeonGreen-KilD (NG-KilD) to track the Kil system assembly on prey (Supplementary Fig. 11), we confirmed that the Tips show varying efficacies in prey recognition, with Tip1 ≥ Tip4 > Tip3 > Tip2 (Fig. 3h).

Overall, these results demonstrate that KilP pilus formation is crucial for *Myxococcus* cells to establish stable contacts with prey cells, facilitating their intoxication. Analysis of single-Tip expressing strains revealed that the different Tips determine the frequency of productive contacts, likely by interacting with prey-specific molecular signals. It is unlikely that the observed differences are due to differences in gene expression because there is no obvious correlation between the expression level of the *tip* genes and Tip efficiency (i.e., the genes encoding Tip1 proteins are expressed at lower levels than Tip3 encoding proteins, Supplementary Fig. 9b).

### Tip1, Tip2, and Tip3 functions require the "needleless" T3SS*

We next explored the functional link between the Kil system and the T3SS*. Both complexes are recruited at the prey contact site, but it is unclear whether this occurs simultaneously. To address this, we tested their colocalization using mCherry-KilD and SctN-eYFP (the T3SS* ATPase) as proxies. We observed synchronized recruitment of these proteins, which could reflect a functional association of both systems during predation (Supplementary Fig. 12).

Importantly, a previous study reported that the assembly of each system on prey is interdependent[28]. In a $\Delta T3SS^*$ strain, it was observed that the prey cells were killed but not-lysed, and no recruitment of the Kil system was detected[28]. This is intriguing, considering that the Kil system is essential for prey killing. In our hands, a $\Delta T3SS^*$ strain ($\Delta mxan\_2438\text{-}2451$) retained the ability to kill its prey on agar plates but failed to lyse it, as demonstrated in a liquid predation assay (Fig. 3a, b and Supplementary Fig. 9a). However, KilP pilus polymerization was still observed in this strain (Fig. 4a), and clusters of NG-KilF (the Kil ATPase) were systematically detected when the prey was killed (Supplementary Fig. 13a, b). Thus, the assembly and the killing function of the Kil system are not dependent on the T3SS* as initially stated[28]. However, the T3SS* cannot localize at the prey contact site in a $\Delta kilACF$ mutant, indicating that Kil assembly further recruits the T3SS* at this site (Supplementary Fig. 13c).

Hence, prey contact involves two distinct steps, a killing step that solely depends on the Kil apparatus and a lysing step that requires the T3SS*. To investigate how the different Tip complexes are involved in this process and how they might functionally connect with the T3SS*, we measured the predation performances of motile strains expressing individual Tips and carrying an additional *T3SS** deletion. First, we observed that Tip1, Tip2, and Tip3 could only kill and lyse prey cells if the T3SS* was present (Fig. 4b and Supplementary Fig. 13d). Confirming these observations, non-motile $\Delta T3SS^*$ strains expressing Tip1, Tip2, or Tip3 failed to kill and lyse *E. coli*, even after 2 h of contact, indicating that the T3SS* is required for the toxic function of these Tips (Fig. 4c, d and Supplementary Fig. 14a). However, it was different in a Tip4-expressing strain, which could still kill preys by contact but failed to lyse these cells in absence of the T3SS* (Fig. 4b–d and Supplementary Figs. 13d, 14a). This indicates that Tip4 works, like the other Tips, in concert with the T3SS* for prey lysis but also carries a T3SS*-independent killing activity. Indeed, deleting both the T3SS* and Tip4 completely abolished prey cell intoxication (Supplementary Fig. 14b–d), demonstrating that the whole toxic activity is carried by the T3SS* (in association with Tip1, 2, and 3) and Tip4 (independently from the T3SS*).

### The prepilin peptidase KilA is linked to the maturation of the Tip2, Tip3, and Tip4 complexes

Previously, we demonstrated that deletion of the *kilA* prepilin peptidase had no effect on the predation efficiency of *Myxococcus*[27]. This was surprising, given that maturation of prepilins into pilins is critical for their incorporation in the polymerizing pilus[4,11,12,15,55]. The absence of a strong deletion phenotype clearly indicates that KilA cannot be responsible for processing KilP. Indeed, KilP polymerization can still be observed in a $\Delta kilA$ strain (Supplementary Fig. 15a). Remarkably, KilA is shorter than canonical prepilin peptidases, which might suggest that this enzyme is no longer functional[56,57]. However, its predicted structure shares strong similarities with *Pseudomonas* FppA, a short prepilin peptidase with a confirmed Flp-processing activity[58] (Supplementary Fig. 15b). Since all the Tip minor pilins contain a predicted prepilin peptidase processing site, KilA might specifically process a subset of these pilins, and the absence of a detectable predation defect in the $\Delta kilA$ strain may be attributed to the redundant and complementary functions of the different Tips.

To test this hypothesis, we deleted *kilA* in single-Tip expressing strains and tested the predation efficiency of each mutant in liquid or on surfaces. We adjusted the prey-to-predator ratio to 4:1, a condition where the predatory performance of a strain expressing Tip2 or Tip3 is easily detectable on agar plates. In this assay, *kilA* was essential for the function of Tip2, Tip3, and Tip4, but not Tip1 (Fig. 5a and Supplementary Figs. 15c, see 15d, e for the 8:1 ratio).

Given that a strain expressing Tip1 phenocopies a $\Delta kilA$ strain in both liquid and solid media (Fig. 5a and Supplementary Fig. 15f), these results strongly suggest that KilA serves as the prepilin peptidase for a

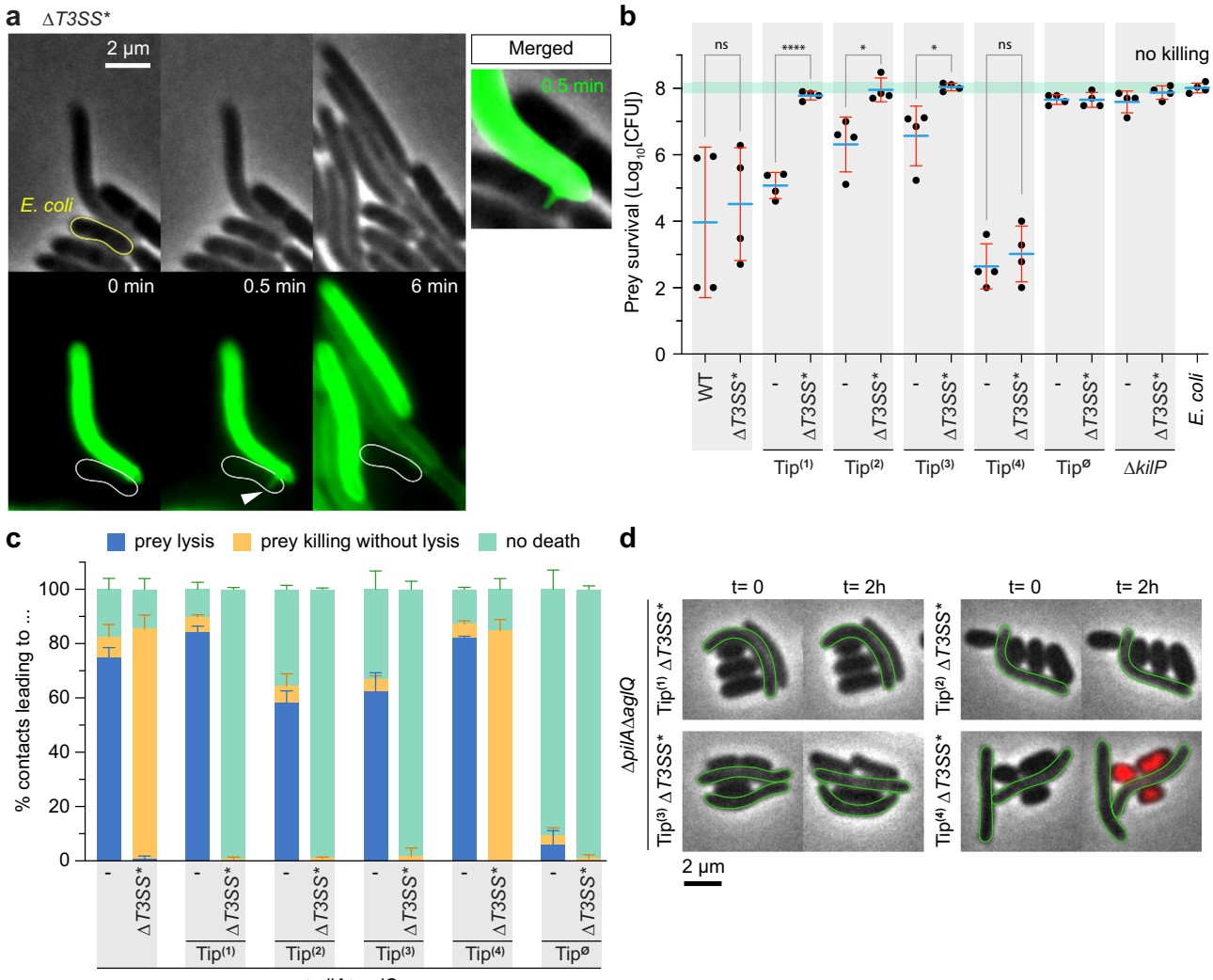

**Fig. 4 | Tip1, Tip2, and Tip3 toxic functions require the "needleless" T3SS*. a** In a motile *ΔT3SS** mutant, polymerization of a KilP[A96C] pilus is still occurring (white arrowhead). **b** Tip1-, Tip2-, and Tip3-expressing strains require a functional T3SS* to kill *E. coli*. In this assay, *E. coli* and motile *Myxococcus* strains were mixed at a 4:1 prey-to-predator ratio and spotted on CF 1.5% agar plates supplemented with 0.075% glucose. After 5 h of incubation, *E. coli* survival upon predation was evaluated by harvesting the corresponding spots for CFU quantification. The prey alone (light green line) was used as control. This experiment was performed independently four times (*n* = 4 per strain). Statistics were determined using a two-tailed unpaired *t*-test. Significance: [ns]$P > 0.05$, *$P < 0.05$, **$P < 0.01$, ***$P < 0.001$, and ****$P < 0.0001$. **c** After 2-h exposure to non-motile *Myxococcus* strains, carrying or not a *T3SS** deletion, we determined by microscopy the percentage of contacts leading to *E. coli* lysis (in blue), *E. coli* death without lysis (in yellow, death was

confirmed with propidium iodide staining), and *E. coli* survival (in pale cyan). The number of prey contacts used for quantification was *n* = 420 for the mutant expressing all Tips, *n* = 476 for *ΔT3SS**, *n* = 548 for Tip[(1)], *n* = 450 for Tip[(1)]*ΔT3SS**, *n* = 463 for Tip[(2)], *n* = 460 for Tip[(2)]*ΔT3SS**, *n* = 497 for Tip[(3)], *n* = 451 for Tip[(3)]*ΔT3SS**, *n* = 470 for Tip[(4)], *n* = 485 for Tip[(4)]*ΔT3SS**, *n* = 582 for Tip[∅], and *n* = 521 for Tip[∅]*ΔT3SS**. Quantifications were performed on three independent experiments. In panels **b**, **c**, error bars represent the standard deviation of the mean. **d** This panel shows representative microscopy images of *E. coli* cells in contact with non-motile single-Tip expressing strains carrying a deletion of the T3SS*. After 2-h incubation, "dead-but-not-lysed" *E. coli* cells are stained in red by propidium iodide (a more detailed figure is available in Supplementary Fig. 14a). Source data and exact *p* values are provided as a Source Data file.

subset of Tips, comprising Tip2, Tip3, and Tip4. Ectopic expression of *kilA* successfully complemented the corresponding deletion strains, confirming the functional association between KilA and these Tips (Supplementary Fig. 16).

**The Flp minor pilins KilK and KilO are functionally associated with the Tip complexes**

Having identified the KilP major pilin and its associated Tip complexes, we next revisited the function of the predicted KilK pilin (*mxan_4655*). The corresponding gene is located in the *kil*-gene cluster II, and its deletion only has a minor effect on predation efficiency[27]. Again, this might indicate a functional redundancy. Interestingly, the *tip3*-gene cluster also contains a gene (*mxan_1150*) encoding a predicted 59-

amino acid long Flp protein referred to as KilO. This protein shares similarities with KilK: it is fully α-helical and features a leader peptide region followed by a conserved cleavage motif (Supplementary Fig. 17a). Consistent with a minor pilin function, *kilK* and *kilO* are transcribed at very low levels (Supplementary Fig. 9b).

In T4P machineries, minor pilin complexes are generally adapted at the extremity of the pilus filament via so-called "coupling/adapter" pilins[18,20,31,59]. We, therefore, investigated whether KilK and KilO had a similar function. Again, a 4:1 prey-to-predator ratio was used to test on agar plates the effect of *kilK* and *kilO* deletions in single-Tip expressing strains (Fig. 5b and Supplementary Fig. 17b). Under this condition, *kilK* or *kilO* deletion mutants effectively killed their prey, although the *ΔkilK* strain already showed a strong predation defect at a 8:1 ratio

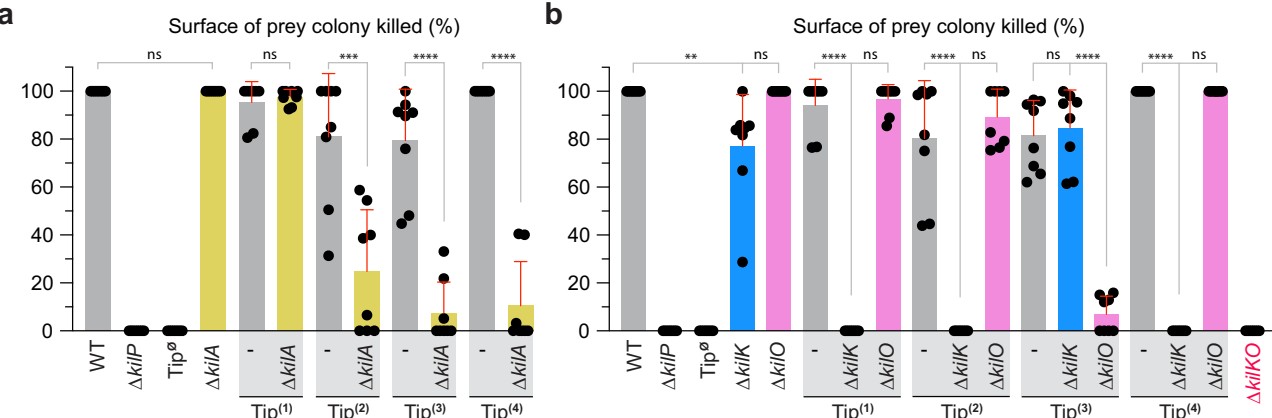

**Fig. 5 | The prepilin peptidase KilA and the minor pilins KilK and KilO are functionalities associated with the Tips. a** This graph represents the predation efficiency on agar plates of motile single-Tip expressing strains carrying a deletion of *kilA* prepilin peptidase. **b** This graph represents the predation efficiency on solid surfaces of motile single-Tip expressing strains carrying a deletion of *kilK* or *kilO* minor pilins. In panels **a**, **b** predation efficiencies were quantified by measuring the surface of *E. coli* colony killed by *Myxococcus* after 24 h of predation. A 4:1 prey-to-predator ratio was used in these assays. All experiments were performed

independently four times. Each data point represents a prey colony (*n* = 8 per strain) grown on a CF 1.5% agar plate supplemented with 0.075% glucose. Error bars represent the standard deviation of the mean. Statistics were determined using two-tailed unpaired *t*-test in (**a**), and ordinary one-way ANOVA followed by Dunnett's multiple comparisons test in (**b**). Significance: $^{ns}P > 0.05$, $^*P < 0.05$, $^{**}P < 0.01$, $^{***}P < 0.001$, and $^{****}P < 0.0001$. Source data and exact *p* values are provided as a Source Data file.

(Supplementary Fig. 17c, d). In the single-Tip expressing strains, a *kilK* deletion abolished the function of Tip1, Tip2, and Tip4. In contrast, the deletion of *kilO* only affected the function of Tip3, which may not be surprising given its association with the *tip3*-gene cluster (Fig. 5b and Supplementary Fig. 17b). Similar conclusions were drawn from liquid predation assays (Supplementary Fig. 17e). Finally, a double *kilKO* deletion strain was incapable of predation in both liquid and solid conditions, phenocopying a Tip$^{\varnothing}$ mutant (Fig. 5b and Supplementary Fig. 17f–h).

If KilK and KilO function as adapters for the pilus Tips, these proteins should be critical to prime KilP polymerization. As expected, a *kilKO* double deletion completely abolished KilP$^{A96C}$ polymerization, phenocopying a mutant lacking all of the Tips (Supplementary Fig. 18a–c). Ectopic expression of *kilK or kilO* successfully complemented the different Δ*kilK or* Δ*kilO* single-Tip expressing strains, further confirming the implication of these proteins in the function of the Tips (Supplementary Fig. 16).

In summary, these findings demonstrate that KilK preferentially works as an "adapter" pilin with Tip1, Tip2, and Tip4, while KilO exclusively works with Tip3. The in vivo expression level of *kilK* is higher than *kilO*, consistent with its association with multiple Tips (Supplementary Fig. 9b).

### A molecular model of the Kil pilus
To understand how the "adapter" pilins and the Tips insert at the pilus extremity, we next built models of the Kil pilus in association with different Tips. These AlphaFold models confidently revealed that the pilus is formed by the self-assembly of the KilP pilin into a left-handed 3-start helix with a diameter of ~46 Å and a helical rise of ~16 Å (Fig. 6a and Supplementary Fig. 19). Remarkably, the C-terminal α2-helix of KilP is unfolded and its charged residues uniformly cover the surface of the pilus, conferring electrostatic properties (Fig. 6b). Overall, all these structural features are consistent with the characteristics of Tad pili[46,49].

These models also revealed interactions between KilP, KilK, KilO, and the Tips. The bundle of α1N helices present at the base of each Tip can perfectly insert at the extremity of the KilP pilus, confirming a "docking" function for this region. In addition, KilK or KilO consistently occupy the same position within the pilus complex: their C-terminal α-helical regions are nestled between KilM, KilN, and KilP, while their

N-terminal α-helical regions are deeply embedded into the KilP pilus (Fig. 6c and Supplementary Figs. 19–21). Thus, the position of KilK and KilO at the interface between the pilus extremity and a Tip is consistent with an adapter function. However, the structural and sequence features that govern KilK and KilO specific association with a given Tip are not explained by these models.

### Predation against diverse prey species has varying Tip requirements
The Kil system is active against evolutionarily distinct bacteria, monoderms as well as diderms[27]. To determine whether this predation versatility is linked to the functional diversity of the Tips, we tested the predation efficiency of single-Tip expressing strains against *C. crescentus*, *B. subtilis* and *E. coli*.

As previously described[27], the wild-type strain efficiently killed all three prey species after 24 h of predation (Fig. 7a and Supplementary Fig. 22). In contrast, the Δ*kilACF*, Δ*kilP*, and Tip$^{\varnothing}$ mutants were all defective in predating all preys, demonstrating that a functional Kil pilus and its associated Tips are essential for predation on a diverse range of bacterial species (Fig. 7a and Supplementary Fig. 22).

Remarkably, single-Tip expressing strains predated with variable efficiencies on distinct prey species, revealing significant functional differences between the Tips. Tip4 was active against all tested prey species, whereas Tip1 showed varying levels of predation efficiency depending on the prey type, following this specific order: *C. crescentus* > *E. coli* > *B. subtilis*. Tip2, which had minimal activity against *E. coli*, showed intermediate efficiency at killing both *C. crescentus* and *B. subtilis*. Tip3 had only a minor effect on *C. crescentus* and no detectable activity against *B. subtilis or E. coli* (Fig. 7a and Supplementary Fig. 22).

These results demonstrate that Tip variation is important not only for predation against a single prey but also to kill different bacterial species. Under our laboratory conditions, it appears that Tip4 is the most efficient Tip, independently of the prey type.

### Tad pili are genetically associated with Tip homologs in other bacterial species
When the Kil system was first characterized, we proposed that it represented a subclass of Tad pili restricted to predatory bacteria, specifically within the *Myxococcales*, *Bdellovibrionales*, and

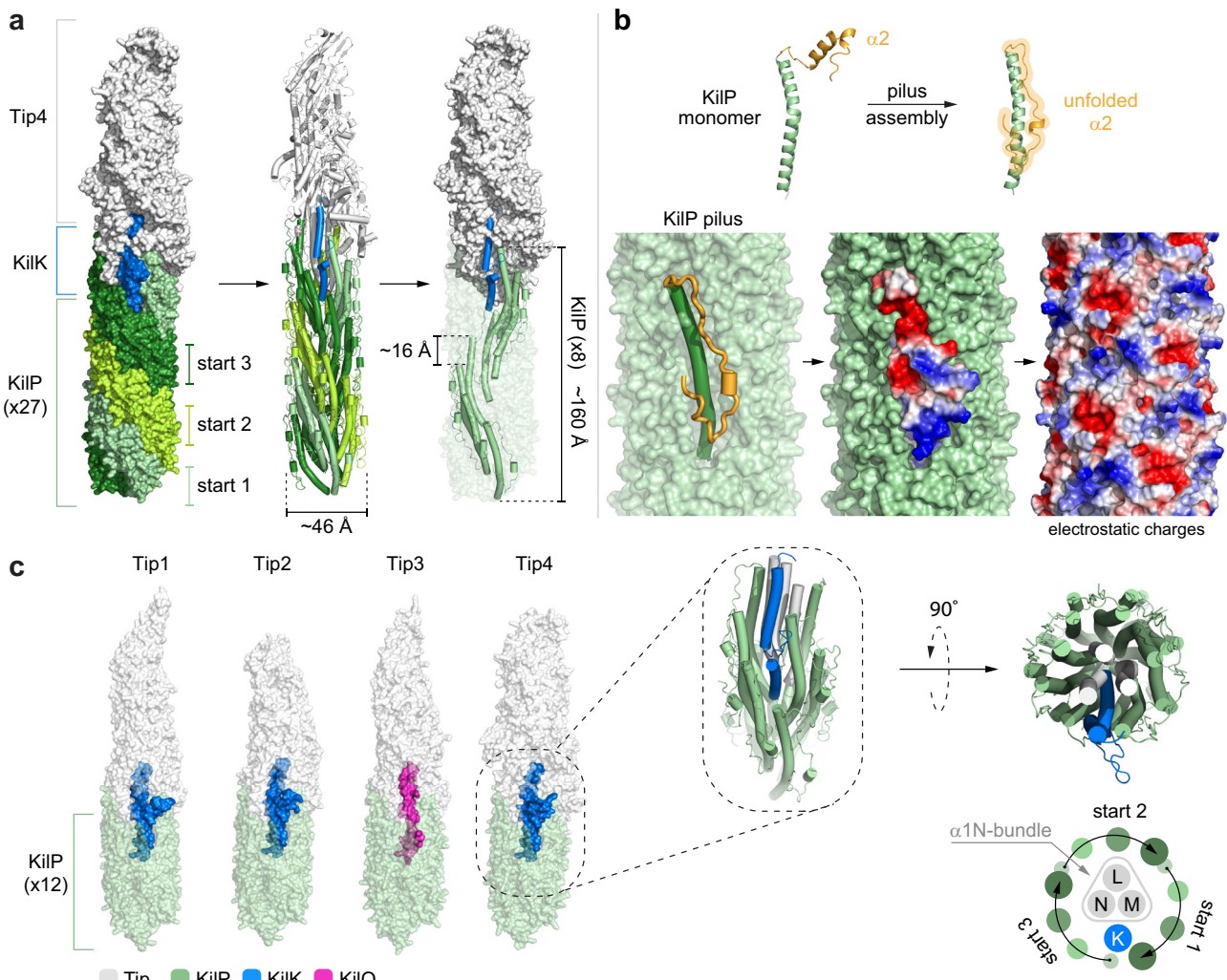

**Fig. 6 | A molecular model of the Kil pilus. a** The KilP pilus is a left-handed three-start helix with a diameter of ~46 Å and a helical rise of ~16 Å. Its extremity is constituted of a heterotrimer of minor pilins (here Tip4) and a Tip adapter (here KilK). **b** Remarkably, the α2 helical region of KilP (highlighted in orange) unfolds within the pilin polymer, exposing charged residues that confer electrostatic surface properties to the pilus (red: negatively charged; blue: positively charged). **c** AlphaFold3 structure predictions of the KilP pilus (in green) interacting with the four different Tips (in light gray) and one "adapter" pilin (KilK in blue or KilO in magenta). The "zoomed-in" view corresponds to the region of interaction between Tip4 and the KilP pilus. For clarity, it only shows the α1N-bundle (in gray) surrounded by multiple copies of KilP (in green) and one copy of KilK (in blue). A cartoon representation of the different α-helices constituting the junction between Tip4 and the pilus extremity is also provided below (see Supplementary Fig. 20 for more details). To generate these AlphaFold predictions, the leader peptide regions were systematically removed from the major and minor pilin amino acid sequences.

*Bradymonadales* orders[26,27,60]. To extend this analysis to the Tips, we analyzed the distribution of Kil-associated minor pilins across various bacterial species. Using the KEGG server[61], we identified orthologous *kil* and *tip* genes in a subset of *Myxococcales* genomes and examined their synteny (Supplementary Fig. 23). This analysis revealed that *kil*- and *tip*-gene clusters are present in these microorganisms, adopting an organization similar to that of *Myxococcus xanthus*.

Next, to further explore the distribution of Kil systems in bacteria, we used a primary subset of genomes (from Supplementary Fig. 23) to generate hidden Markov model (HMM) profiles for the most conserved *kil* or *tip* genes (Supplementary Data 1). These HMM profiles were further used to investigate the distribution of Kil-associated minor pilins across all bacterial species for which a complete genome was available. At least one homolog of a *kil* gene was identified in 21,258 genomes out of 38,791, spanning 47 bacterial phyla out of 64 (Supplementary Fig. 24a and Supplementary Data 1). This first analysis revealed that the Kil-associated *tip1*, *tip2*, *tip3*, and *tip4* genes are

restricted to *Myxococcota* genomes and almost exclusively in the *Myxococcales* (Supplementary Fig. 24b and Supplementary Data 1). Thus, the four Tips constitute the majority of the *tip*-gene repertoire, suggesting that this diversity is sufficient for the *Myxococcales* to interact with the prey species encountered in their respective ecological niches.

However, it is possible that sequence diversity hampers the discovery of new predatory Tips. To search for other possible Tip-like complexes, we, therefore, employed a structural homology approach to identify distant KilN-like analogs using the Foldseek[62] and KEGG servers. This analysis revealed a fifth type of Tip in some *Myxococcales* and importantly, it also permitted to identify potential novel Tip complexes in other predatory bacteria, including *Bradymonas sediminis* and *Bdellovibrio bacteriovorus* (Fig. 7b and Supplementary Figs. 23, 25). Despite a strong resemblance, these Tips presented notable structural variations, thereby expending the repertoire of Tips. This structural diversity may reflect adaptation to a different prey range or indicate an alternative function.

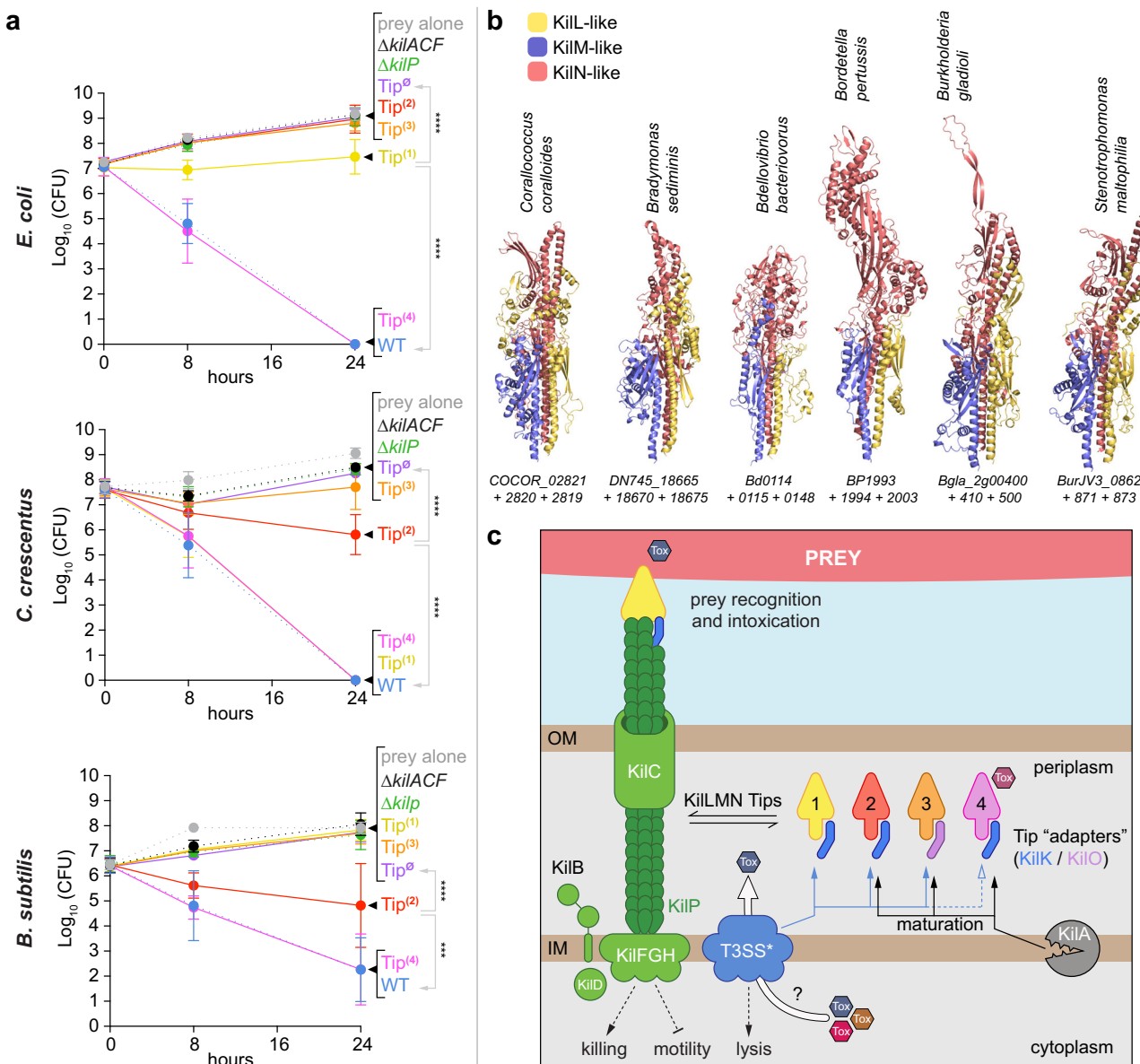

**Fig. 7 | Predation against diverse prey species has varying Tip requirements. a** These different graphs show the survival of *E. coli*, *C. crescentus*, and *B. subtilis* after 0, 8, and 24 h of predation in the presence of motile single-Tip expressing strains. In this assay, the prey and predator cells were mixed to an 8:1 ratio and spotted on CF 1.5% agar plates supplemented with 0.075% glucose. Prey survival during predation was evaluated over time by harvesting the predation spots for CFU quantification. The prey alone (gray circles) was used as control. Predation efficiencies were also evaluated for the WT, Δ*kilACF*, Δ*kilP*, and Tip^Ø strains. This experiment was independently performed five times. Error bars represent the standard deviation to the mean. Statistics were determined using ordinary one-way ANOVA followed by Dunnett's multiple comparisons test. Significance compared to WT or Tip^Ø after 24 h of predation: ***$P < 0.001$ and ****$P < 0.0001$. **b** Tip-like complexes are also present in other bacterial species with predicted Tad pili, including predatory bacteria (*C. coralloides*, *B. sediminis*, and *B. bacteriovorus*) and pathogens (*B. pertussis* Tohama I, *B. gladioli* BSR3 and *S. maltophilia* JV3). The gene numbers corresponding to the proteins constituting these Tips are indicated below the AlphaFold predictions. See Supplementary Figs. 25, 26 for a detailed structural analysis of the Tip complexes described in this study. To generate these AlphaFold predictions, the leader peptide regions were systematically removed from the minor pilin amino acid sequences. **c** Cartoon representation of a KilP pilus polymerizing at the prey contact site. Our data suggest that the extremity of the KilP pilus is equipped with four different Tips, each positioned at the pilus end by a specific adapter (KilK or KilO). Maturation of Tip2, Tip3, and Tip4 involves the prepilin peptidase KilA, while Tip1 and KilP are probably processed by a different and unknown prepilin peptidase. Our current model proposes that, upon prey detection, *Myxococcus* pauses and assembles the Kil system along with the "needleless" T3SS* to form a predation complex involved in the transport and delivery of Tip-specific toxic effectors, leading to prey killing and lysis. Source data and exact *p* values are provided as a Source Data file.

Expanding our investigations beyond predatory bacteria, we also identified structural analogs of the Tip complexes in human pathogenic bacteria, including *Pseudomonas aeruginosa* PAO1, *Bordetella pertussis* Tohama I, *Burkholderia gladioli* BSR3 or *Stenotrophomonas maltophilia* JV3. In these microorganisms, all the *tad*-encoding regions contain *tadE*, *tadF* and *tadG* genes[33,34,63,64] and AlphaFold predicts that the corresponding minor pilins assemble into dimers or trimers, structurally resembling the predation Tips (Fig. 7b and Supplementary Fig. 26). In these models, the large TadG minor pilin constitutes the backbone of the complex and presents a C-terminal modular domain, similar to the KilN protein. These findings strongly suggest that the extremity of Tad pili in other bacteria is also functionally customized by "Tip-like" complexes of minor pilins.

## Discussion

Contact-dependent killing is the major mechanism underlying *Myxococcus* predation. This process involves a complex series of events, in which contact and prey detection trigger a signaling cascade that leads to motility arrest, prey cell lysis and ultimately, to motility resumption[27,65]. At the prey contact site, two molecular machineries are assembled: a Tad-like system (the Kil) and an atypical secretion system (T3SS*). These systems are essential at each step of the predation process, including prey detection, motility arrest, killing, and lysis[27,28,30]. How each machinery contributes to this process is poorly understood.

In this work, we show that the Kil machinery polymerizes a pilus upon contact with prey cells. In most cases, KilP pili are shorter than the prey width, but if they extend further, they curve upon reaching the opposite side of the prey cell. This indicates that these filaments interact tightly with the prey, either by penetrating or wrapping around it.

How does the Kil pilus drive predation? Our findings demonstrate that the Kil pilus is essential for prey detection and killing, with genetic evidence suggesting that these functions are mediated by interchangeable Tip complexes present at the extremity of the KilP pilus. Specifically, this study reveals that:

i. The Tips have partially redundant functions in predation, and deleting all four Tips is required to obtain a Δ*kilP*-like phenotype. This clearly indicates that the entire set of minor pilins associated with the KilP pilus is encoded within these four gene clusters (Fig. 7c).

ii. In T4Ps, minor pilins are generally essential for pilus extension, as they prime its polymerization[17–19,59,66]. Consistent with this, any of the Tip complexes is sufficient for pilus polymerization, but deleting all four Tips fully abolishes pilus formation.

iii. Minor pilins can only become functional if processed by a prepilin peptidase[4,12,55]. Our genetic analysis demonstrates that the prepilin peptidase KilA is essential for the maturation of Tip2, Tip3, and Tip4 but not Tip1 or KilP (Fig. 7c). This finding explains the moderate predation phenotype observed for a Δ*kilA* mutant in our original study[27]. KilP and Tip1 are likely processed by another prepilin peptidase. Prepilin peptidases are indeed known for their substrate promiscuity[12,16,67–69]. It is, therefore, possible that the T4aP prepilin peptidase PilD (*mxan_5779*) acts as a general prepilin peptidase, capable of processing a subset of unrelated pilins. However, this hypothesis cannot currently be tested, as *pilD* is essential and cannot be deleted[56,57].

iv. In T4Ps, "adapter" pilins are essential for coupling a minor pilin complex to a pilus extremity[18,20,31,59]. We identified not just one, but two "adapter" pilins, KilK and KilO, critical for the KilP pilus function and its Tips. This explains the Δ*kilK* intermediate predation phenotype reported in our original study[27]. Deleting both genes led to a Tip^∅-like phenotype, with no pilus polymerization events detected in this strain. Thus, KilK and KilO constitute the complete set of adapters and are probably essential for positioning the Tips at the pilus extremity, priming its polymerization. This hypothesis is strongly supported by our AlphaFold models, which systematically position a Tip complex at the pilus extremity, with its α1N-bundle deeply embedded into the KilP filament. In these models, KilK or KilO are consistently positioned at the same location within the pilus, interacting with the Tip's α1N-bundle and KilP subunits. However, these models do not reveal any distinctive features that would explain the specific association of these Flps with a given Tip.

What is the function of the Tips? Our genetic analysis reveals that the ability to pause, assemble the Kil system, polymerize a pilus, and kill the prey is Tip specific. We also show that strains expressing individual Tips present variable predation efficiencies depending on the

prey type. This suggests that the Tips play a crucial role in prey recognition. The prey recognition capability of the Tips probably resides in the exposed and variable domain of the larger KilN pilin, which may detect specific molecular signals on the prey cell surface. Although a single-Tip is sufficient for predation, all Tips perform redundant functions in wild-type cells. This redundancy suggests that the Kil system might select specific Tip combinations depending on the prey species. Such a combinatorial strategy could explain how myxobacteria prey on a wide variety of microorganisms. Tip activity also varies depending on the conditions (e.g., solid vs liquid environments), highlighting the influence of the ecological niches and environmental contexts on Tip functionality and/or selection. However, the mechanism by which the Kil system selects a specific Tip remains unclear. It is also unknown whether the KilP pilus consists of a single filament or a bundle of filaments, as observed in some T4P systems[12,42,45,58,70]. It is, therefore, possible that multiple KilP pili are simultaneously equipped with different Tips, thereby increasing predation efficiency. Hence, to determine the pilus composition and architecture during predation, particularly in the presence of different prey species, Cryo-EM studies are the logical next step.

The pilus Tips are also important for prey cell intoxication and function together with the "needleless" T3SS* (Fig. 7c). The recruitment of the T3SS* is Kil-dependent, and both systems simultaneously co-localize, suggesting that they assemble to form a larger complex. Importantly, the Kil system alone is not sufficient to acquire nutrients from the prey, as it can induce prey cell death via Tip4 but lacks the ability to lyse the prey, a process dependent on the T3SS*. How could these two systems function together? The *Myxococcus* T3SS* is incomplete and consists of a cytoplasmic sorting platform associated with an inner membrane export apparatus, both normally required for translocating toxic effectors from the cytoplasm into the needle complex[28,30,35,71,72]. Thus, the Tips could substitute for the missing needle complex and act as vehicles for the delivery of T3SS*-associated effectors into target cells. However, the functional redundancy makes it difficult to disentangle de various Tip functions. Since all four Tips are required for the T3SS*-dependent lytic effect, they could all deliver the T3SS* effectors individually, acting maybe like a T2SS. Remarkably, Tip4 carries an additional toxic activity, which is sufficient on its own to puncture and kill prey cells without lysis. This suggests that Tip4 also associates with a T3SS*-independent toxin or carries itself a toxic activity.

In summary, these results uncover a unique association between a Tad-like pilus and a "needleless" T3SS. These systems probably coordinate during predation, enabling prey recognition, killing, and lysis via the delivery of predation effectors, perhaps channeled by the Kil pilus and its associated Tips, similar to a T2SS[24,25,73].

The analysis of diverse bacterial genomes revealed that Tip-like minor pilins are restricted to predatory myxobacteria, suggesting that their functions have specifically evolved for contact-dependent killing. Beyond myxobacteria, these proteins could not be identified through primary amino acid sequence alignments. However, structural analogs were detected in other predatory bacteria with Kil-like systems. Whether these systems mediate prey recognition and killing in these microorganisms remains unknown. Nonetheless, genetic evidence suggests that a Kil-like apparatus plays a crucial role in prey invasion by *Bdellovibrio bacteriovorus*, an endobiotic predator that penetrates prey cells and develops within their periplasm[26,74]. Hence, investigating the function of the predicted *Bdellovibrio* Tip complex, particularly during the prey penetration phase, would provide valuable insights.

Beyond predatory bacteria, the discovery of the Kil Tips reveals, for the first time, that functional diversification of Tad pili is driven by minor pilin complexes exposing variable domains at their extremities, similar to what has been described in some T4P subclasses[18–24,31,32,75–78]. Remarkably, phylogenetically distant bacterial species with *tad-like* loci consistently carry genes encoding TadEFG-like minor pilins (see

references[26,33,34,60,63,64] for comprehensive *tad* locus maps). These minor pilins are required for pilus formation and function[33,34,42,79−82]. Thus, we propose that, in these Tad machineries, TadEFG minor pilins also form a Tip complex, which not only initiates pilus polymerization but also confers functional properties to the pilus. For instance, the pathogen *Vibrio vulnificus* contains three distinct *tad* loci and deletion of all three is required to observe a strong virulence defect[34,83]. This suggests that these Tad pili have specialized and partially redundant functions in virulence, likely relying on the associated TadEFG complexes. Further characterization of Tip-like proteins in bacteria will be essential to elucidate the functional repertoire of Tad systems.

## Methods

### Strain construction
Supplemental information on plasmids and primers used in this study are provided in Supplementary Data 2. Strain information can be found in Supplementary Data 3.

CYE liquid medium (10 mM MOPS−pH 7.6, 1% Bacto-casitone, 0.5% yeast extract, and 0.04 mM $MgSO_4$) was used to grow the different *Myxococcus* strains. CYE agar plates were prepared by adding 1.5% of Bacto-agar to the medium.

The different *M. xanthus DZ2* deletion and fluorescent strains were generated using a double crossover recombination strategy, as previously described[27]. Briefly, ~800-base pair fragments corresponding to the 5′ and 3′ regions flanking the gene of interest were used for allelic exchange by recombination. For this, the different PCR fragments were Gibson assembled (Thermo Fisher Scientific) into the pBJ114 plasmid. pBJ114 is a suicide plasmid carrying the *nptI* gene for initial selection on kanamycin and the *galK* gene for counter-selection on galactose. The plasmids were then transformed in *E. coli* DH5a and plated on LB agar plates supplemented with 50 μg/ml of kanamycin. After purification and sequencing, the plasmids were transformed into *Myxococcus* by electroporation. Single crossover integrants were selected on CYE plates supplemented with 100 μg/ml of kanamycin. Counter-selection for the second crossover event was carried out by growing kanamycin-resistant colonies in liquid CYE under nonselective conditions. The next day, serially diluted culture samples were plated on CYE agar petri dishes supplemented with 2.5% galactose. Kanamycin-sensitive colonies containing the exchanged allele were identified by PCR screening.

To fluorescently tag the T3SS* ATPase SctN with the yellow fluorescent protein (eYFP), we used the pEYFP integrative plasmid. A fragment of ~800 base pairs corresponding to the 3′ region of the gene of interest (without its stop codon) was cloned in frame with *eyfp*. After sequencing, the purified plasmid was electroporated in *Myxococcus* and transformants were selected on CYE agar plates supplemented with 100 μg/ml of kanamycin.

To complement the deletion strains, the different genes of interest (*kilP, kilA, kilK,* or *kilO*) were ectopically and constitutively expressed under the control of the *pilA* promoter (*mxan_5783*). For this, the wild-type alleles of these genes were cloned into a pSWU19 plasmid carrying the *pilA* promoter region (~1000 bp). After electroporation of the different deletion strains, transformants with the integrated plasmid at the Mx8 *att* attachment site were selected on CYE agar plates supplemented with 100 μg/ml of kanamycin. Strains transformed with the empty pSWU19-PpilA vector were used as controls.

Similarly, to fluorescently label the KilP pilus, a *kilP^A96C* mutant allele (carrying a substitution of Ala96 to a cysteine) was cloned into the pSWU19-PpilA plasmid for ectopic and constitutive expression.

### Predation assays in liquid and on solid surfaces
Predation assays in liquid and agar plates were performed as previously described[27].

*E. coli* MG1655 and *B. subtilis* were grown in liquid LB (BD Difco) and *C. crescentus* NA1000 was grown in liquid PYE (0.2% peptone, 0.1% yeast extract, 1 mM $MgSO_4$, and 0.5 mM $CaCl_2$). Agar plates were prepared by adding 1.5% of Bacto-agar to the medium.

Liquid CF starvation medium (10 mM MOPS−pH 7.6, 1 mM $KH_2PO_4$, 8 mM $MgSO_4$, 0.02% $(NH_4)_2SO_4$, 0.015% Bacto-casitone, and 0.2% sodium citrate) was used to perform the different predation assays. CF agar plates were prepared by adding 1.5% of Bacto-agar to the medium.

All the predator or prey cultures were grown at 32 °C.

Briefly, for predation assays on solid surfaces, overnight CYE cultures of the different *Myxococcus* strains and LB cultures of *E. coli* MG1655 were pelleted and resuspended in liquid CF to a final $OD_{600}$ of 5. Depending on the prey-to-predator ratio used, 200 μl of *E. coli* cells was either mixed with 25 μl (8:1 ratio), 50 μl (4:1 ratio), or 100 μl (2:1 ratio) of *Myxococcus* cells in a 96-well plate. A 10-μl aliquot of these cell suspensions was then spotted onto CF 1.5% agar plates supplemented with 0.075% glucose. Glucose allows minimal growth of the prey, but it cannot be metabolized by *Myxococcus*. These agar plates were next incubated at 32 °C for 24 h, and predation spots were imaged using a DFK 38UX267 camera (Imaging Source) mounted on an Olympus SZ61 stereo microscope. The IC Capture 2.5 software (Imaging Source) was used for image acquisition. Images were analyzed with the Fiji software (ImageJ 2.14.0/1.54 f) to measure the area of the prey colony killed by *Myxococcus*[84].

The complementation of a Δ*kilA* or Δ*kilK* Tip2-expressing strain was evaluated using a similar protocol but with *C. crescentus* as prey.

*Myxococcus* predation efficiency was evaluated in liquid by measuring the hydrolysis kinetics of ChloroPhenol Red-β-D-Galactopyranoside (CPRG, Roche). In this colorimetric assay, CPRG hydrolysis is catalyzed by the β-Galactosidase released in the supernatant by lysed *E. coli* during predation. Briefly, an overnight LB culture of *E. coli* MG1655 supplemented with 100 μM of Isopropyl β-D-1-thiogalactopyranoside (IPTG, GoldBio) was pelleted by centrifugation and resuspended in liquid CF to a final $OD_{600}$ of 10. Similarly, overnight cultures of the different *Myxococcus* strains were pelleted and resuspended in liquid CF to a final $OD_{600}$ of 0.5. After supplementation with 100 μM of IPTG, 100 μl of the *E. coli* cell suspension was mixed with 100 μl of *Myxococcus* cells in a 96-well plate. The plate was sealed with breathable tape and incubated at 32 °C in an orbital shaker at 160 rpm. After 24 h, 10 μl of the supernatant was mixed with 140 μl of Z-buffer (60 mM $Na_2HPO_4$, 40 mM $NaH_2PO_4$, 10 mM KCl, and 1 mM $MgSO_4$) supplemented with 20 μg/ml of the CPRG substrate in a 96-well plate. After incubation at 37 °C, the β-galactosidase enzymatic reaction was stopped by adding 65 μl of 1 M $Na_2CO_3$. Absorbances at 576 nm were measured with a TECAN Spark plate reader using the SparkControl V3.1 SP1 acquisition software. To convert the absorbances to Miller units, the different values were normalized by the incubation time and the volume of supernatant used in the enzymatic reaction.

### Prey CFU quantification
To quantify the number of prey cells that survived predation, colony-forming units (CFUs) were determined as previously described[27]. Briefly, kanamycin-resistant strains of *E. coli* MG1655, *B. subtilis*, or *C. crescentus* NA1000 were cultured overnight in the appropriate liquid medium, pelleted, and resuspended in liquid CF to an $OD_{600}$ of 5. Depending on the prey-to-predator ratio used, 200 μl of these prey suspensions were mixed in a 96-well plate with either 25 μl (8:1 ratio) or 50 μl (4:1 ratio) of *Myxococcus* WT or mutant cells resuspended in liquid CF to an $OD_{600}$ of 5. A 10-μl aliquot of this mix was then spotted on CF 1.5% agar plates supplemented with 0.075% glucose and incubated at 32 °C. At 0-, 5-, 8- or 24-h post-incubation, spots were harvested using an inoculation loop and resuspended in 500 μl of liquid CF supplemented with 10 μg/ml of kanamycin to kill *Myxococcus*. These cell suspensions were serially diluted in a 96-well plate, and 5 μl of each

10-fold dilution was spotted on LB agar plates (supplemented with 10 μg/ml of kanamycin) for *E. coli* and *B. subtilis* or on PYE agar plates (supplemented with 25 μg/ml of kanamycin) for *C. crescentus*. After 24-h incubation at 32 °C, CFUs were determined and the number of prey cells that survived within the predation spot was calculated. Images of the prey colonies were captured as described above.

## Microscopy

Detailed information on the microscopy parameters used in this study can be found in Supplementary Data 4. All the movies and images were acquired at 24 °C.

The different *Myxococcus* strains and prey species were grown overnight in the appropriate liquid media. One milliliter of each culture was centrifuged at 600×*g* for 5 min at room temperature. The cell pellets were then resuspended in liquid CF (supplemented with 1 mM CaCl$_2$) to an OD$_{600}$ of ~12 for the prey and ~6 for *Myxococcus*. Equal volumes of predator and prey cell suspensions were mixed together, and 1.5 μl was spotted on a freshly prepared CF 1.5% agar pad supplemented with 1 mM CaCl$_2$. All the pads for microscopy were prepared using a Gene Frame adhesive system (0.25 mm, Thermo Fisher Scientific) and pasted on a glass slide to prevent sample evaporation. Once the spot had dried, the pad was covered with a glass coverslip. Microscope slides prepared with motile strains were incubated in the dark at room temperature for 20–30 min prior to imaging, allowing *Myxococcus* cells to resume motility on the pad. In contrast, slides prepared with non-motile strains were imaged immediately without incubation.

To perform microscopy of *ΔT3SS** strains, 30 μM of propidium iodide (Thermo Fisher Scientific) was added to the CF agar pad to stain in red "dead-but not-lysed" prey cells during predation.

Time-lapse microscopy was performed using an inverted epi-fluorescence Nikon Eclipse Ti2 microscope equipped with a ×100/1.4 DLL objective and a Teledyne Photometrics Kinetix camera. This microscope also features a "Perfect Focus System" (PFS), which automatically maintains focus, even in the presence of mechanical or thermal perturbations. Typically, 30-min movies were recorded, capturing both phase-contrast and fluorescent images every 30 s. To minimize photo-bleaching and phototoxicity, appropriate filters were used, and exposure times were kept to a minimum. Images were acquired using NIS software from Nikon (NIS-Elements AR 5.42.06).

Image analysis and quantification were performed with Fiji software[84].

## KilP pilus labeling

To label the KilP pilus, 100 μl of an overnight CYE culture (supplemented with kanamycin at 100 μg/ml) of the different *Myxococcus* strains carrying the pSWU19-*PpilA::kilP*$^{A196C}$ plasmid was mixed with 2 μl of AlexaFluor-488-C$_5$ maleimide (1 mg/ml in DMSO, Thermo Fisher Scientific). After incubation in the dark at room temperature for 20 min, cells were centrifuged at 600×*g* for 5 min at room temperature. The pellet was then resuspended in 200 μl of liquid CF supplemented with 1 mM CaCl$_2$ and centrifuged again at 600×*g* for 5 min. Finally, depending on its size, the cell pellet was resuspended in 5 to 10 μl of CF supplemented with 1 mM CaCl$_2$. Prior imaging, cells were left in the dark at room temperature for 30 min. Microscope slides were prepared as described above by mixing *Myxococcus* cells with its prey. The slides were imaged after 20–30 min of incubation at 24 °C, except for the non-motile strains, which were imaged immediately after the spot had dried on the pad.

Image analysis, quantification, and measurements were performed using Fiji software[84].

## AlphaFold structure predictions and structure analyses

To generate the different structure predictions of the pilins, the leader peptide regions were systematically removed from their amino acid sequences. The different models were generated using ColabFold and AlphaFold3 with default settings[40,41,85]. The highest-confidence prediction models were used to make structural alignments, measurements, and figures. All AlphaFold models generated in this study are available in Supplementary Data 5.

The structure models for KilP (https://alphafold.ebi.ac.uk/entry/Q1D248; accessed November 19, 2022), FppA (https://alphafold.ebi.ac.uk/entry/A0A1Y0GHK3; accessed March 10, 2024), and KilA (https://alphafold.ebi.ac.uk/entry/Q1D7R5; accessed March 10, 2024) were downloaded from the AlphaFold protein structure database[48].

Structure visualization was performed with ChimeraX 1.7[86] and PyMOL (The PyMOL Molecular Graphics System, Version 3.0.3, Schrödinger, LLC).

Protein-protein contact maps were generated using MAPIYA contact map server[87] (https://mapiya.lcbio.pl/) with a contact cutoff distance of 4 Å.

To identify structural analogs of the Tips, we submitted the AlphaFold models to the DALI server[88] (http://ekhidna2.biocenter.helsinki.fi/dali/) and the Foldseek search server[62] (https://search.foldseek.com/search). Additionally, the DALI "All Against All Structure Comparison" tool was used to determine the Tip equivalence of the AlphaFold-predicted structures presented in Supplementary Fig. 25.

## RNAseq analysis

A pre-existing RNAseq library was used to determine the read counts of the different *kil-pilin* genes[89]. Raw reads have been deposited in the NCBI Sequence Read Archive (SRA) under the Bioproject accession number PRJNA1040215[89]. Briefly, the count matrix was normalized by gene length to calculate transcripts per million (TPM), enabling the comparison of expression levels across genes.

## Search for *kil* and *tip* orthologs in bacteria

A four-step approach was used to explore the diversity of the *kil* and *tip* genes found in *Myxococcus xanthus* DK1622 (CP000113.1) across other species. First, *kil* and *tip* homologs were identified in the KEGG database[61] (https://www.kegg.jp/kegg/) based on sequence similarity. Second, the classification was refined using Foldseek[62], by aligning structures against large protein structure databases. This step identified homologous systems in various predatory bacteria, in which 22 *kil* and *tip* genes and their synteny were conserved. Third, a hidden Markov model (HMM) profile was built with HMMER v3.4[90] (http://hmmer.org) for each of the genes from the multiple sequence alignments (MSA) performed with MAFFT v7.490 (L-INS-i method)[91]. The generated files are provided in Supplementary Data 1. These 22 HMM profiles were used to explore the diversity of homologs in the most recent complete genome assemblies (publicly available in April 2024). For this, the list of 38,791 latest complete genome assemblies referenced in the GTDB taxonomy[92] was used with the NCBI Datasets command-line tools (CLI) ncbi-datasets-cli-16.19.0 to download from the NCBI RefSeq database all coding sequences in the corresponding assemblies. Then, the coding sequences were translated using a Python script with Biopython v 1.78 (https://github.com/biopython/), and PyHMMER 0.8.2[93] was used to extract the best hits with a minimal E-value of $10^{-25}$. The same Python script was finally used to create a presence / absence matrix for each genome and HMM profile. Genomic coordinates being indexed, we used gene indexes to explore gene synteny of the best hits. Finally, a fourth step consisted in going a little bit further to the limit of sequence homology by looking for other Tip-like pilins in few other bacterial species harboring a Tad-*like* pilus using AlphaFold3 predictions[41], Foldseek structure comparison[62] and KEGG ortholog/paralog search[61].

## Amino acid sequence alignments

Protein sequences were aligned using the Uniprot Align server (https://www.uniprot.org/align)[94].

## Statistical analyses

Statistical analyses were performed using GraphPad Prism 10.3.1 software.

## Reporting summary

Further information on research design is available in the Nature Portfolio Reporting Summary linked to this article.

## Data availability

The RNA-seq datasets used in this study come from raw transcriptome sequencing data (consisting of raw reads) which were previously deposited in the NCBI Sequence Read Archive (SRA) under the Bioproject accession number PRJNA1040215[89]. All AlphaFold models generated in this study are available in Supplementary Data 5. Source data are provided with this paper.

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

## Acknowledgements

We thank Dr Philippe Ortet (CEA, BIAM) for the initial help in generating HMM profiles. We thank Dr Hung Le Nguyen for the gift of the pBJ114-ΔT3SS* plasmid. We thank Dr Dorothée Murat, Dr Emilia Mauriello, Dr Romain Mercier, and Dr Vladimir Pelicic for their helpful discussions. This work was supported by the European Union (European Research Council (ERC), grant JAWS885145), the Centre National de la Recherche Scientifique (CNRS, France), Aix-Marseille University (amU, France), and the Institut de Microbiologie, Bioénergies et Biotechnologie (IM2B, funding Emergence—Nouveaux Entrants).

## Author contributions

Conceptualization and writing—original draft: J.H. and T.M.; Experimental work: J.H., L.M., C.L.M., M.B., R.J., and E.M.; Visualization: J.H.; Formal analysis: J.H., C.L.M., and T.M.; Resources and funding acquisition: T.M.; Writing—review and editing: J.H., C.L.M., and T.M.

## Competing interests

The authors declare no competing interests.
