## [Transparent Peer Review file · Nature Communications]

Tad-pili with adaptable tips mediate contact-dependent killing during bacterial predation

Corresponding Author: Dr Tam Mignot

Version 0:

Reviewer comments:

Reviewer #1

(Remarks to the Author)

To the authors:

In this manuscript, the authors present a very interesting study focusing on the subunits comprising the filament of the recently discovered “kil” system in *Myxococcus* that is used for bacterial predation. The topic of the work is of broad interest and relevant to multiple research communities. The authors identify the major pilin of the “kil” tad pilus-like complex as “KilP” which they are able to label and demonstrate is produced after contact with prey cells. The authors identify and characterize the role of minor pilin tip complexes in prey-cell interactions and cell killing, and AlphaFold modeling support the hypothesis that specific tip complexes mediate killing of specific prey species. While the data shown are robust and generally support the main conclusions of the paper, I believe some simple additional experiments are needed to exclude alternative hypotheses as detailed below:

Major comments:

- The authors imply that KilP filaments penetrate *E. coli* as the KilP filament appears to bend and not extend past the *E. coli* cell boundary. From the data presented it looks like it is possible that this filament could bind to the outside of the *E. coli* cell, and perhaps mechanically rotate it during pilus extension leading to similar bending profiles. The S1 movie provided does not refute this possibility, as the *E. coli* cells clearly move in response to KilP production. It would be nice to see 3D data demonstrating the filament actually penetrates the cell to paint a clearer picture on the mechanism of killing.
- The authors draw conclusions that the different tip complexes mediate species-specific prey killing, and while the data generally support this model, it is unclear how different tip mutations affect kil pilus production and how this might also affect species-specific killing. Since tip complexes are generally essential for pilus production, it is possible that the significant defects in killing by Tip1/4 mutants are not due to a lack of those tips *per se*, but is more due to defective kil pilus assembly. It would be nice to see data on kil pilus assembly when only each a single Tip is present (1-4) and in the Tip0 mutant to determine whether defects in assembly may contribute to defects in killing to exclude this possibility.
- The section about KilK and KilQ acting as minor pilins, while intriguing, is not supported by experimental data. The main suggestion that the KilK and KilQ proteins are adapter minor pilins is through AlphaFold modeling – the authors could do simple biochemistry experiments to qualitatively compare the abundance of KilK and KilQ present in cells compared to KilP and the tip complexes with which they associate – if these flp pilins are true minor pilins, they should have low abundance compared to the major pilin KilP and similar abundance to the Tip complexes they are thought to associate with. RNAseq data may also be able to suggest this, though assessment of protein levels would be more conclusive.
- The authors should provide the confidence data for the models presented in Fig S21

Minor comments:

- Why did the authors skip “KilN” as a protein name? The newly named minor pilins jump from L and M to O which seems odd

Reviewer #2

(Remarks to the Author)

The manuscript by Herrou et al. describes how the Myxococcus Kil apparatus and a T3SS basal body synergize during predation. Using extensive genetic analyses, the authors demonstrate that the Kil apparatus assembles a pilus fiber that incorporates distinct Tip complexes to distinguish and target various prey species. Remarkably, Tip function is linked to the assembly of a unique T3SS basal body complex that does not appear to elaborate a needle structure (T3SS*), raising the compelling hypothesis that the Tip complex plays an important role in T3SS* effector delivery. Using AlphaFold modeling, the authors propose pilus tip assembly mechanisms. Additional bioinformatics approaches identified related Tad pilus assemblies (kil- and tip-orthologous components) in phylogenetically diverse bacteria that are potentially decorated with minor pilin complexes harboring variable domains, highlighting the interchangeability of multiple secretion and pilus systems. This study builds on previous reports demonstrating a role of T3SS* in prey cell lysis and Kil-mediated predation. Overall, the mechanistic and biological conclusions of the study were largely supported by the data. However, concerns regarding the proposed Kil/T3SS* killing mechanism/model remain. Addressing the following points would help to strengthen the findings and improve the manuscript.

Major concerns:

1. In line 184, the authors conclude that KilP fibers have the capacity to wrap around the prey bacterium and penetrate the prey envelope when the fiber length exceeds the cell width. However, the presented microscopy data cannot definitively demonstrate that the pilus penetrates the bacterium – an alternative hypothesis is that the pilus remains on the prey cell surface and wraps around the prey outer membrane without penetrating into the periplasm or cytoplasm. Additional microscopy or biochemical evidence supporting the claim that the Kil pilus enters either the prey periplasm or cytoplasm should be provided.
2. In experiments presented in Fig. 3A, the authors conclude that the Tip order of importance for solid surface killing is Tip4>Tip1>Tip3>Tip2 (line 293). However, in Fig. 3D and 3E, the authors demonstrate that prey cell lysis resulting from contact-dependent killing occurs in all cases regardless of the integrated Tip protein. The authors go on to show that WT and Tip4 strains do not require the T3SS* for prey cell lysis (Fig. 4B). It appears from the data presented in Fig. 3E that prey cells are lysed by non-motile strains regardless of the Tip pilin subtype. Studies using the double and single Tip knockouts should be used to show that prey cell lysis ultimately requires T3SS* and specific Tip proteins in the non-motile background.
3. The authors should test the hypothesis presented in line 568 describing a role of T4aP prepilin peptidase in Tip1 and KilP maturation.
4. Likewise, the authors should perform studies to show that a T3SS* mutant expressing Tip2 or Tip 3 alone would be unable to kill and lyse prey as outlined in lines 603 – 604.
5. Can the authors speculate as to how T3SS* effectors are delivered to target cells? One intriguing possibility is that Kil pili mediate intimate predator-prey contacts that facilitate T3SS* pore formation and effector delivery through membrane fusion or some alternative mechanism that does not require T3SS pilus biogenesis (perhaps similar to a T6SS that loads various effectors onto the javelin tip for direct delivery into host cells?).
6. Can the authors demonstrate Kil apparatus and T3SS* structural interactions in the periplasm or at the predator-prey contact site via microscopy or other biochemical approaches? These data would significantly strengthen the model presented in Fig. 6C.
7. Similarly, can the authors propose a model for Tip loading? Is it more likely to be stochastic, randomly assembled and bundled pili, or hierarchical loading based on cellular protein levels? Given the data demonstrating a hierarchy of Tip functionality in solid surface and liquid culture prey killing, Tip loading may occur in a preferential manner that is regulated by external cues.

Version 1:

Reviewer comments:

Reviewer #1

(Remarks to the Author)

The authors have satisfactorily addressed all of my concerns.

Reviewer #2

(Remarks to the Author)

The revised manuscript by Herrou et al. is substantially improved by the inclusion of new data and text clarifications. Importantly, the authors performed multiple additional experiments to demonstrate co-localization of the T3SS* and Kil system (a major advance over the previous manuscript), new evidence supporting the roles of the minor pilins as Tip adaptors, functional assays that incorporate genetically-complemented strains, and additional experiments to demonstrate Tip association with the T3SS* machinery which is required for prey lysis when associated with Tips 1-3). This reviewer appreciates the extensive text modifications that tone down previously over-interpreted data and clarifies data presentation. Overall, the revised manuscript fills multiple important gaps in both experimental rigor and model refinement, leading to an improved study that significantly advances the field. Future proposed studies using cryo-ET will hopefully provide additional insight into the molecular mechanisms underlying T3SS*/Kil-mediated prey killing and lysis.

RESPONSE TO REVIEWERS' COMMENTS

Tad-pili with adaptable tips mediate contact-dependent killing during bacterial predation

Julien Herrou, Laetitia My, Caroline Monteil, Marine Bergot, Rikesk Jain, Emmanuelle Martinez, Tâm Mignot.

In response to the reviewers' comments, we have significantly revised the manuscript and incorporated substantial new data. The main updates are highlighted in blue, and we have renamed the Kil pilin proteins as suggested by one of the reviewers (details provided below). In this revised version, we demonstrate that the T3SS* and the Kil system colocalize at the prey contact site. Furthermore, we show that the recruitment of T3SS* is Kil-dependent, while the reciprocal is not true. Predation assays and quantitative analyses clearly indicate that the predatory function of a subset of Tips is directly linked to the T3SS*. Additionally, we provide new evidence supporting the roles of the minor pilins KilK and KilO (previously referred to as KilQ) as Tip adaptors. The main text has also been revised to offer a more nuanced interpretation of the interaction between the predation pilus and the prey. As detailed in our response to reviewers, accurately resolving this interaction remains a challenging issue, which will be addressed in future studies.

Additional updates included in the manuscript are as follows:

- We now provide complementation data for all strains carrying deletions of *kilA*, *kilK*, or *kilO*.
- The cutoff in the interaction maps between KilL, KilM, and KilN has been refined. Previously, we used the default settings of the MAPIYA server (cutoff 8 Å), which was overly generous for studying residue interactions. The updated maps now use a stricter 4 Å cutoff.
- All source data are now provided in a comprehensive Source Data file.

Reviewer #1 (Remarks to the Author):

To the authors

In this manuscript, the authors present a very interesting study focusing on the subunits comprising the filament of the recently discovered "kil" system in *Myxococcus* that is used for bacterial predation. The topic of the work is of broad interest and relevant to multiple research communities. The authors identify the major pilin of the "kil" tad pilus-like complex as "KilP" which they are able to label and demonstrate is produced after contact with prey cells. The authors identify and characterize the role of minor pilin tip complexes in prey-cell interactions and cell killing, and AlphaFold modeling support the hypothesis that specific tip complexes mediate killing of specific prey species. While the data shown are robust and generally support the main conclusions of the paper, I believe some simple additional experiments are needed to exclude alternative hypotheses as detailed below:

We thank Reviewer #1 for the generally positive evaluation our manuscript and providing valuable comments and suggestions.

A detailed point-by-point response to Reviewer #1's comments is provided below.

Major comments

1. The authors imply that KilP filaments penetrate *E. coli* as the KilP filament appears to bend and not extend past the *E. coli* cell boundary. From the data presented it looks like it is possible that this filament could bind to the outside of the *E. coli* cell, and perhaps mechanically rotate it during pilus extension leading to similar bending profiles. The S1 movie provided does not refute this possibility, as the *E. coli* cells clearly move in response to KilP production. It would be nice to see 3D data demonstrating the filament actually penetrates the cell to paint a clearer picture on the mechanism of killing.

The reviewer's point is valid. To date, all our attempts to determine whether the Kil pilus extends inside or outside the prey by microscopy have been inconclusive. After testing 3D-SIM approaches, we concluded that Cryo-Electron Tomography (Cryo-ET) is the only available method with the potential to determine whether the Kil pilus is extending inside the prey cell. However, this approach is a long term goal and, we are currently working on predation conditions that would allow predator-prey interactions to occur directly on an EM grid. Detecting the assembled Kil pilus via Cryo-ET would first require localization of the complex using Cryo-fluorescence microscopy. Additionally, Cryo-FIB milling would likely be necessary to reduce the sample's thickness before proceeding with Cryo-ET acquisition and reconstruction. Given the substantial technical demands and time required for this approach, it will be pursued as part of a future research project.

To address reviewer's comment, we have modified the text for a more cautious interpretation, toning down the conclusion that KilP fibers penetrate prey cells.

Result section:

"Occasionally, the pilus exceeded the prey cell width (~0.8 μm), and in these cases, it systematically curved when reaching the opposite side of the prey cell. This suggests that long KilP fibers either wrap around or more likely, penetrate the prey, bending when it reaches the cell membrane on the opposite side".

Discussion section:

"In this work, we show that the Kil machinery polymerizes a pilus in contact with prey cells and the majority of the observed pilin filaments are short. However, when longer pili are detected, they curve as they reach the opposite side of the prey cell. This demonstrates that these filaments interact tightly with the prey, either by penetrating its cell envelope or by wrapping around it."

2. The authors draw conclusions that the different tip complexes mediate species-specific prey killing, and while the data generally support this model, it is unclear how different tip mutations affect kil pilus production and how this might also affect species-specific killing. Since tip complexes are generally essential for pilus production, it is possible that the significant defects in killing by Tip1/4 mutants are not due to a lack of those tips per se, but is more due to defective kil pilus assembly. It would be nice to see data on kil pilus assembly

when only each a single Tip is present (1-4) and in the Tip0 mutant to determine whether defects in assembly may contribute to defects in killing to exclude this possibility.

This is a good point. This data was in fact already included in the manuscript, but the way we presented it was unclear. Figure 3d precisely shows that single Tip-expressing strains can all support pilus polymerization but not a strain lacking all Tips (Tip⁰). We also find that the frequency of pilus assembly is Tip-dependent, suggesting that more efficient Tips promote more frequent killing contacts.

We have revised the main text to clarify this:

“In these single-Tip expressing strains, we characterized the contact-dependent killing of *E. coli* and determined that prey lysis efficiency, corresponding to the percentage of contacts leading to *E. coli* lysis, was Tip-dependent. Tip performances for prey lysis followed the same trend observed on agar plates, with Tip1 ≥ Tip4 > Tip3 > Tip2 (Fig. 3c). Additionally, pilus polymerization was observed in all these strains, except in a mutant lacking all four Tips (Tip⁰) (Fig. 3d). Thus, each of the Tip complexes is sufficient to prime pilus extension, which only becomes abolished when all Tips are deleted. Upon prey contact, the frequency of pilus polymerization also varied depending on the Tip, following the same trend as above, with Tip1 ≥ Tip4 > Tip3 > Tip2 (Fig. 3d). Importantly, when a pilus was formed, it was systematically associated with prey lysis in all these strains (Fig. 3e). In conclusion, the Tips show varying efficacies in prey cell recognition, but they all support killing.”

3. The section about KilK and KilQ acting as minor pilins, while intriguing, is not supported by experimental data. The main suggestion that the KilK and KilQ proteins are adapter minor pilins is through AlphaFold modeling – the authors could do simple biochemistry experiments to qualitatively compare the abundance of KilK and KilQ present in cells compared to KilP and the tip complexes with which they associate – if these flp pilins are true minor pilins, they should have low abundance compared to the major pilin KilP and similar abundance to the Tip complexes they are thought to associate with. RNAseq data may also be able to suggest this, though assessment of protein levels would be more conclusive.

Another valid point. Despite our attempts, detecting the proteins by western blot is proving challenging, likely because of the low expression levels. We therefore have added RNAseq experiments reporting the transcription level values of *kilK* and *kilO* (formerly *kilQ*, see Reviewer’s #2 minor comment #5) to the graph in Supplementary Figure 9b. As expected for minor-pilin-encoding genes, expression levels of *kilK* and *kilO* are very low (much lower than KilP). Remarkably, the expression level of *kilK* is slightly higher than *kilO*, which may not be surprising given that KilK is genetically associated to multiple Tips (Tip1, Tip2 and Tip4), while KilO is only associated to Tip3.

To further validate the hypothesis that KilK and KilQ are essential for adapting the Tips at the pilus end, we quantified the frequency of pilus polymerization in a strain deleted for both *kilK* and *kilO* (see Supplementary Figure 18). Since the Tips are critical to prime KilP polymerization, a double *kilKO* deletion should completely abolish pilus polymerization if KilK and KilO truly function as Tip “adapters”. Consistent with this, no pilus polymerization was

detected in a $\Delta kilKO$ strain, phenocopying a “no-Tip” mutant. This demonstrates that KilK and KilO are minor pilins essential for the assembly of the Kil pilus.

Finally, the ectopic and constitutive expression of *kilK* or *kilO* successfully complemented the predation defect observed in the single-Tip expressing strains with deletions of *kilK* or *kilO* (see Supplementary Figure 16). We can therefore conclude that these pilins are essential for the function of the Tips.

In summary, we provide a range of evidence - including Tip-specific predation defects, structural predictions, transcription levels, and pilus polymerization defect - suggesting that KilK and KilO are minor pilins essential for Tip-dependent pilus function.

4. The authors should provide the confidence data for the models presented in Fig S21

We added confidence data for the models presented in Supplementary Figure 25 (formerly Fig. S21) and Supplementary Figure 26 (formerly Fig. S22).

Minor comments

5. Why did the authors skip “KilN” as a protein name? The newly named minor pilins jump from L and M to O which seems odd

We agree, it does seem odd. We also realized that we inadvertently inverted the names of *mxan_4658* and *mxan_4660* in the manuscript. In our first study, *mxan_4658* was referred to as *kilL* and *mxan_4660* as *kilM*. In agreement with previous annotations and to take under consideration the reviewer’s suggestion, we have renamed the proteins as follow (changes are highlighted in red in the table):

Gene number	Before change	After change
mxan_4655	KilK	KilK
mxan_4658	KilM	KilL
mxan_4660	KilL	KilM
mxan_4659	KilO	KilN
mxan_1150	KilQ	KilO
mxan_5121	KilP	KilP

Reviewer #2 (Remarks to the Author):

The manuscript by Herrou et al. describes how the Myxococcus Kil apparatus and a T3SS basal body synergize during predation. Using extensive genetic analyses, the authors demonstrate that the Kil apparatus assembles a pilus fiber that incorporates distinct Tip complexes to distinguish and target various prey species. Remarkably, Tip function is linked to the assembly of a unique T3SS basal body complex that does not appear to elaborate a needle structure (T3SS*), raising the compelling hypothesis that the Tip complex plays an important role in T3SS* effector delivery. Using AlphaFold modeling, the authors propose pilus tip assembly mechanisms. Additional bioinformatics approaches identified related Tad pilus assemblies (kil- and tip-orthologous components) in phylogenetically diverse bacteria that are potentially decorated with minor pilin complexes harboring variable domains, highlighting the interchangeability of multiple secretion and pilus systems. This study builds on previous reports demonstrating a role of T3SS* in prey cell lysis and Kil-mediated predation. Overall, the mechanistic and biological conclusions of the study were largely supported by the data. However, concerns regarding the proposed Kil/T3SS* killing mechanism/model remain. Addressing the following points would help to strengthen the findings and improve the manuscript.

We thank Reviewer #2 for carefully reviewing our manuscript and providing useful remarks and suggestions.

A detailed point-by-point response to Reviewer's #2 comments is provided below.

Major concerns:

1. In line 184, the authors conclude that KilP fibers have the capacity to wrap around the prey bacterium and penetrate the prey envelope when the fiber length exceeds the cell width. However, the presented microscopy data cannot definitively demonstrate that the pilus penetrates the bacterium – an alternative hypothesis is that the pilus remains on the prey cell surface and wraps around the prey outer membrane without penetrating into the periplasm or cytoplasm. Additional microscopy or biochemical evidence supporting the claim that the Kil pilus enters either the prey periplasm or cytoplasm should be provided.

This point is also raised by Reviewer #1. Please see our detailed answer. In summary, both reviewers are right, we are currently unable to determine whether the Kil pilus penetrates the prey or wraps around it in a definitive manner. We are currently exploring long term Cryo-Electron Tomography (Cryo-ET) approaches to resolve this point. We changed the text to tone down the conclusion that the pilus penetrates the prey cell.

2. In experiments presented in Fig. 3A, the authors conclude that the Tip order of importance for solid surface killing is Tip4>Tip1>Tip3>Tip2 (line 293). However, in Fig. 3D and 3E, the authors demonstrate that prey cell lysis resulting from contact-dependent killing occurs in all cases regardless of the integrated Tip protein. The authors go on to show that WT and Tip4 strains do not require the T3SS* for prey cell lysis (Fig. 4B). It appears from the data presented in Fig. 3E that prey cells are lysed by non-motile strains regardless of the Tip pilin subtype.

Studies using the double and single Tip knockouts should be used to show that prey cell lysis ultimately requires T3SS* and specific Tip proteins in the non-motile background.

Based on this comment, it seems that our presentation of the role of the T3SS* and its connection with the Tips needed clarification. The critical aspect is that prey killing and lysis can be genetically uncoupled due to the existence of a lysing activity that depends on the T3SS*(in association with each of the Tips) and a Tip4 killing activity (without lysis) that is independent of the T3SS*.

We clarify this in the revision and added additional experiments showing that:

1. Single Tip1, Tip2 and Tip3 expressing strains can no longer kill and lyse prey cells if the T3SS* is deleted, demonstrating their functional association with the T3SS* (Figure 4b-d and Supplementary Figure 14a).
2. Tip4-expressing strains can kill and lyse prey cells but, while prey killing is T3SS*-independent (as measured by propidium iodide staining), prey lysis is dependent on the T3SS* (Figure 4b-d and associated Supplementary Figure 14a).
3. Deletion of both T3SS* and Tip4 (Tip1, Tip2, and Tip3 are still present) leads to absence of prey killing and lysis in motile, but also in non-motile strains even after 2 hours of forced contact (Supplementary Figure 14b-d).

Together, these results demonstrate that the Kil Tips function in association with the T3SS*. While Tip1, Tip2 and Tip3 are **strictly** associated with the T3SS*, Tip4 is an exception, as it exhibits a T3SS*-independent killing activity but requires the T3SS* for prey cell lysis. This suggests that all Tips may accommodate T3SS* toxic cargoes and Tip4 additionally mediates a T3SS*-independent toxic activity, either directly or indirectly. We modified the text extensively to clarify the interpretation.

3. The authors should test the hypothesis presented in line 568 describing a role of T4aP prepilin peptidase in Tip1 and KilP maturation.

This is a good suggestion. Unfortunately, the T4aP prepilin peptidase PilD is essential and cannot be deleted. To clarify this, we added the following sentence to the manuscript:

“It is therefore possible that the T4aP prepilin peptidase PilD (*mxan_5779*) acts as a general prepilin peptidase, capable of processing a subset of unrelated pilins. However, this hypothesis cannot currently be tested, as *pilD* is essential and cannot be deleted (Friedrich C. *et al.*, Konovalova A. *et al.*)”.

Friedrich C, Bulyha I, Sogaard-Andersen L. 2014. Outside-In Assembly Pathway of the Type IV Pilus System in *Myxococcus xanthus*. *Journal of Bacteriology* 196:378–390.

Konovalova A, Petters T, Sogaard-Andersen L. 2010. Extracellular biology of *Myxococcus xanthus*. *FEMS Microbiol Rev* 34:89–106.

4. Likewise, the authors should perform studies to show that a T3SS* mutant expressing Tip2 or Tip 3 alone would be unable to kill and lyse prey as outlined in lines 603 – 604.

As requested, we generated non-motile $\Delta T3SS^*$ strains expressing Tip2 or Tip3 individually. Similar to the $\Delta T3SS^*$ Tip1-expressing strain, these strains failed to kill and lyse preys. The corresponding results are reported in Figure 4c,d and associated Supplementary Figure 14a.

5. Can the authors speculate as to how T3SS* effectors are delivered to target cells? One intriguing possibility is that Kil pili mediate intimate predator-prey contacts that facilitate T3SS* pore formation and effector delivery through membrane fusion or some alternative mechanism that does not require T3SS pilus biogenesis (perhaps similar to a T6SS that loads various effectors onto the javelin tip for direct delivery into host cells?).

The main difficulty for T3SS* effector delivery is that the T3SS* lacks components that would allow crossing of the OM. At this stage, we favor the hypothesis that the Kil Tips deliver T3SS* cargoes, similar to a T2SS. In the discussion, we propose this modified paragraph:

“The pilus Tips are also important for prey cell intoxication and function together with the “needleless” T3SS* (Fig. 7c). The recruitment of the T3SS* is Kil-dependent and both systems co-localize simultaneously, suggesting that they assemble to form a larger complex. Importantly, the Kil system alone is not sufficient to acquire nutrients from prey, as it can induce prey cell death *via* Tip4 but lacks the ability to lyse the prey cell, a process dependent on the T3SS*. How could these two systems function together? The *Myxococcus* T3SS* is incomplete and is composed of a cytoplasmic sorting platform associated to an inner membrane export apparatus, both required for translocating toxic effectors from the cytoplasm into the needle complex^{28,30,35,71,72}. Thus, the Tips could substitute for the needle complex and act as vehicles for the delivery of T3SS*-associated effectors into target cells. However, the functional redundancy makes it difficult to disentangle de various Tip functions. Since all four Tips are required for the T3SS*-dependent lytic effect, they could all deliver the T3SS* effectors individually, acting maybe like a T2SS. Remarkably, Tip4-carries an additional toxic activity, which is sufficient on its own to puncture and kill prey cells. This suggests that Tip4 also associates with a T3SS-independent toxin or carries itself a toxic activity.

In summary, these results uncover a unique association between a Tad-like pilus and a needleless T3SS. These systems probably coordinate during predation, enabling prey recognition, killing and lysis via the delivery of predation effectors, perhaps channeled by the Kil pilus and its associated Tips, similar to a T2SS^{24,25,73}.”

6. Can the authors demonstrate Kil apparatus and T3SS* structural interactions in the periplasm or at the predator-prey contact site via microscopy or other biochemical approaches? These data would significantly strengthen the model presented in Fig. 6C.

In the manuscript, we now provide fluorescence microscopy evidence demonstrating that the Kil system and the T3SS* colocalize at the prey contact site (Supplementary Figure 12), and that the recruitment of the T3SS* is Kil-dependent (Supplementary Figure 13c). However, the reverse is not true, as the T3SS* is not required for the recruitment of the Kil system (Supplementary Figure 13a,b). Overall, this suggests a pathway where prey contact first promotes Kil assembly, which then recruits the T3SS* directly or indirectly. At this stage, we have not identified any proteins involved in this interaction. This task is an important direction for future research but will require extensive explorations given that both systems include numerous proteins of unknown functions.

7. Similarity, can the authors propose a model for Tip loading? Is it more likely to be stochastic, randomly assembled and bundled pili, or hierarchical loading based on cellular protein levels?

Given the data demonstrating a hierarchy of Tip functionality in solid surface and liquid culture prey killing, Tip loading may occur in a preferential manner that is regulated by external cues.

This is an excellent question. We do not currently have a model, given that we do not know if indeed the pili are organized as single fibers or bundles. Since we see functional hierarchies, we can imagine that Tip loading may also be influenced by the ecological niches and environmental conditions. In the discussion, we write:

“Although a single Tip is sufficient for predation, all Tips perform redundant functions in wild-type cells. This redundancy suggests that the Kil system might select specific Tip combinations depending on the prey species. Such a combinatorial strategy could explain how myxobacteria prey on a wide variety of organisms. Tip activity also varies depending on the conditions (e.g., solid vs liquid environments), highlighting the influence of the ecological niches and environmental contexts on Tip functionality and/or selection. The mechanism by which the Kil system selects a specific Tip remains unclear. It is also unknown whether the KilP pilus consists of a single filament or a bundle of filaments, as observed in some T4P systems^{12,42,45,58,70}. It is therefore possible that multiple KilP pili are simultaneously equipped with different Tips, thereby increasing predation efficiency. Cryo-EM studies are the logical next step to determine the pilus composition and architecture during predation, particularly in the presence of different prey species.”

RESPONSE TO REVIEWERS' COMMENTS

Tad-pili with adaptable tips mediate contact-dependent killing during bacterial predation

Julien Herrou, Laetitia My, Caroline Monteil, Marine Bergot, Rikesk Jain, Emmanuelle Martinez, Tâm Mignot.

We thank both reviewers for the careful evaluation of this manuscript and for providing insightful recommendations that significantly improved this work.

Reviewer #1 (Remarks to the Author):

The authors have satisfactorily addressed all of my concerns.

Reviewer #2 (Remarks to the Author):

The revised manuscript by Herrou *et al.* is substantially improved by the inclusion of new data and text clarifications. Importantly, the authors performed multiple additional experiments to demonstrate co-localization of the T3SS* and Kil system (a major advance over the previous manuscript), new evidence supporting the roles of the minor pilins as Tip adaptors, functional assays that incorporate genetically-complemented strains, and additional experiments to demonstrate Tip association with the T3SS* machinery which is required for prey lysis when associated with Tips 1-3). This reviewer appreciates the extensive text modifications that tone down previously over-interpreted data and clarifies data presentation. Overall, the revised manuscript fills multiple important gaps in both experimental rigor and model refinement, leading to an improved study that significantly advances the field. Future proposed studies using cryo-ET will hopefully provide additional insight into the molecular mechanisms underlying T3SS*/Kil-mediated prey killing and lysis.